# Mitigating Forgetting in LLM Supervised Fine-Tuning and Preference Learning

## Abstract

Post-training of pre-trained LLMs, which typically consists of the supervised fine-tuning (SFT) stage and the preference learning (RLHF or DPO) stage, is crucial to effective and safe LLM applications. The widely adopted approach in post-training popular open-source LLMs is to sequentially perform SFT and RLHF/DPO. However, sequential training is sub-optimal in terms of SFT and RLHF/DPO trade-off: the LLM gradually forgets about the first stage's training when undergoing the second stage's training. We theoretically prove the sub-optimality of sequential post-training. Furthermore, we propose a practical joint post-training framework that has theoretical convergence guarantees and empirically outperforms sequential post-training framework, while having similar computational cost.

## 1 Introduction

Recent years have witnessed the great capabilities of large language models (LLMs) trained on a large corpus of datasets (OpenAI, 2022; Dubey et al., 2024; Abdin et al., 2024). These models have been applied to a wide range of tasks including virtual assistant (OpenAI, 2022), code development (Roziere et al., 2023), and education/research (Achiam et al., 2023). Typically LLMs undergo the pre-training phase and the post-training phase. The post-training phase adapts the pre-trained LLM to specific tasks, thus is crucial to its successful applications.

The post-training phase of LLMs often has two stages (Abdin et al., 2024; Dubey et al., 2024): the Supervised Fine-Tuning (SFT) stage and the preference learning stage. Typical methods for preference learning include Reinforcement Learning from Human Feedback (RLHF) (Ouyang et al., 2022), and Direct Preference Optimization (DPO) (Rafailov et al., 2024). Given this two-stage process, a natural approach is to performing *sequential training*, e.g., first perform DPO then SFT or vice verse. For example, the instruct variant of popular open-source models like PHI-3 (Abdin et al., 2024) or LLAMA-3 (Dubey et al., 2024) sequentially undergo SFT and DPO training. Or in other scenarios like continual learning of an aligned model, the process can be interpreted as sequentially performing DPO/RLHF followed by SFT (Tang et al., 2020; Qi et al., 2023; Fang et al., 2024).

However, sequential training of RLHF and SFT is sub-optimal in terms of the trade-off between preference learning and SFT. When the model is undergoing the second stage of training, it gradually and inevitably *forgets about* the first stage's training. In this case, we argue that even regularization like KL divergence used in RLHF/DPO cannot avoid forgetting due to the data distribution shift from the SFT dataset to the preference dataset. An illustration of the sub-optimality of sequential post-training is shown in Figure 1 (left), where we observe that sequential training leads to the increase of the DPO objective during SFT, resulting in a worse trade-off between the two objectives than the method to be introduced in this work. Similar issue has also been observed in, e.g., (Qi et al., 2023). To overcome this issue and achieve a better trade-off, a naive thought is to mix the preference objective and SFT objective by minimizing their linear scalarization. However, the naive mixing method is computationally inefficient in practice, since the optimization objectives are different with different formats of data. The increased computational complexity can be observed in Figure 1 (right), where mixing significantly increases the cost. The cost is prohibitively high in LLM training due to the size of the model. Therefore, in this work, we aim to answer the following question:

*Can we design a post-training framework that achieves better trade-off than the sequential training method, while having reduced computational cost than naive mixing?*

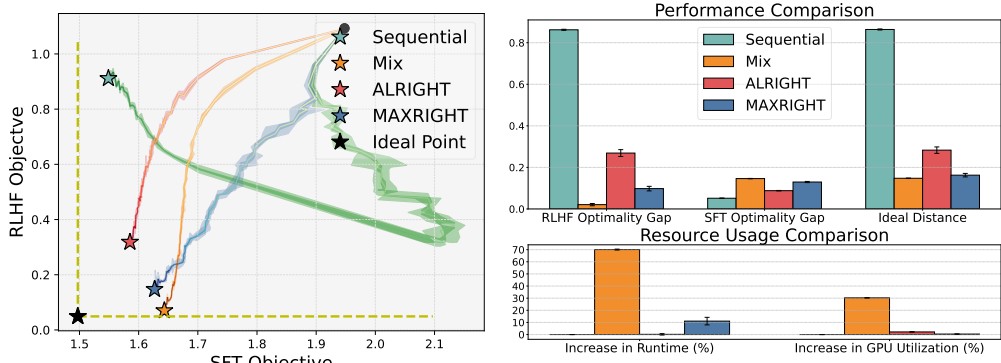

Figure 1: **Efficient Trade-off in RLHF and SFT Optimization.** Sequential optimization, commonly used to align and fine-tune pre-trained models, often biases the model towards the last objective it was optimized on, as illustrated by the optimization trajectories in the objective space (left) and the performance comparison (top right, lower the better). In contrast, simultaneous optimization of a **Mix** of RLHF and SFT objectives achieves a more balanced performance but requires significantly more resources (bottom right, lower the better). We propose **ALRIGHT** and **MAXRIGHT** methods for simultaneous RLHF and SFT optimization, offering an improved trade-off with minimal extra cost.

**Our contributions.** To this end, we propose a joint SFT and DPO training framework with the ALRIGHT and MAXRIGHT variants. The contributions of this work can be summarized as follows:

C1) **Insights into the forgetting issue of two-stage sequential training.** We provide theoretical results on the forgetting issue of sequential method, which is further supported by empirical evidence. Specifically, we prove that sequentially doing DPO and SFT can have a non-diminishing optimality gap. To our best knowledge, this is the first theoretical result on the suboptimality of sequentially learning the SFT and DPO objectives. We further conduct experiments on LLAMA-3-8B and PYTHIA-1B, empirical results support our claims.

C2) **Principled post-training methods with almost no extra cost.** We propose post-training algorithms with theoretical guarantees, which outperform the sequential method while enjoying lower computational cost than the mixing method. Specifically, we propose 1) ALternating supeRvised fIne-tuninG and Human preference alignmenT (ALRIGHT) method, which provably converges to any desired trade-off between DPO and SFT objectives; and 2) MAXimum supeRvised fIne-tuninG and Human preference alignmenT (MAXRIGHT) method, which adaptively alternates between optimizing RLHF and SFT objectives.

C3) **Strong empirical performance on standard benchmarks.** Using the LLAMA3-8B model, our methods outperform the sequential approach by up to 3% on the MMLU (1-shot) benchmark (Hendrycks et al., 2020) and achieve up to a 31% increase in win rate on the RLHF dataset (evaluated by GPT-4-TURBO), with minimal additional computation.

**Technical challenges.** The main technical challenge lies in theoretically proving the issue of forgetting when optimizing SFT and DPO losses (two log-likelihood functions). Existing lower bounds for continual learning in *non-LLM* settings using SGD rely on quadratic objectives (Ding et al., 2024), where iterate updates are more tractable due to the linear structure of the gradient.

However, the problem we address involves negative log-likelihood objectives, whose gradients are non-linear with respect to the parameters. This non-linearity complicates the analysis, requiring careful construction of an example that is guaranteed to exhibit undesirable performance. Additionally, we must analyze the behavior of the updated parameters throughout both stages of optimization and derive conditions that relate this behavior to the lower bound of the performance. We successfully overcome these challenges, and the details of this analysis are provided in Appendix A.2.

## 2 PRELIMINARIES

In this section, we formally introduce the notations and problem setup for DPO and SFT.

*Model.* We denote the LLM parameter to be optimized for either RLHF or SFT by $\theta$, and we use $\pi_\theta(y \mid x)$ to denote the LLM that generates an output $y$ given an input $x$ for an SFT or RLHF task.

*DPO.* We consider using DPO (Rafailov et al., 2024) to align $\theta$ with a preference dataset, given by $\mathcal{D}_{\text{DPO}} = \{x_{\text{DPO}}^{(i)}, y_w^{(i)}, y_\ell^{(i)}\}_{i=1}^{N_1}$, where $N_1$ is the number of data, $x^{(i)}$ are the inputs for the LLM, $y_w^{(i)}$ and $y_\ell^{(i)}$ are the chosen (preferred) and rejected (dispreferred) responses to $x^{(i)}$, respectively, for all $i \in \{1, \ldots, N_1\}$. The DPO objective is given by

$$f_{\text{DPO}}(\theta; \mathcal{D}_{\text{DPO}}, \pi_{\text{ref}}, \beta) := -\frac{1}{N_1} \sum_{x_{\text{DPO}}, y_w, y_\ell \in \mathcal{D}_{\text{DPO}}} \left[ \log \left( \sigma \left( \beta \log \left( \frac{\pi_\theta(y_w \mid x_{\text{DPO}})}{\pi_{\text{ref}}(y_w \mid x_{\text{DPO}})} \right) - \beta \log \left( \frac{\pi_\theta(y_\ell \mid x_{\text{DPO}})}{\pi_{\text{ref}}(y_\ell \mid x_{\text{DPO}})} \right) \right) \right) \right],$$
(1)

where $\sigma$ is the sigmoid function, $\pi_{\text{ref}}$ is a given reference model, and $\beta$ is a regularization constant that penalize the objective when $\pi_\theta(y \mid x)$ is diverging too much from $\pi_{\text{ref}}(y \mid x)$ on $x \sim \mathcal{D}_{\text{DPO}}$. In sequential training, when DPO is performed before SFT, we use the model trained on the chosen responses in $\mathcal{D}_{\text{DPO}}$ as $\pi_{\text{ref}}$. When SFT is performed before DPO, the model obtained after the SFT phase is used as the $\pi_{\text{ref}}$. Given a data point $(x_{\text{DPO}}, y_w, y_\ell)$, the gradient estimate of $f_{\text{DPO}}$ is given as

$$g_{\text{DPO}}(\theta; x_{\text{DPO}}, y_w, y_\ell, \pi_{\text{ref}}, \beta) := -\left(1 - \sigma(h_\beta(\theta; x, y_w, y_\ell, \pi_{\text{ref}}))\right) \nabla_\theta h_\beta(\theta; x_{\text{DPO}}, y_w, y_\ell, \pi_{\text{ref}}), \quad (2)$$

where

$$h_\beta(\theta; x_{\text{DPO}}, y_w, y_\ell, \pi_{\text{ref}}) := \beta \log \left( \frac{\pi_\theta(y_w \mid x_{\text{DPO}})}{\pi_{\text{ref}}(y_w \mid x_{\text{DPO}})} \right) - \beta \log \left( \frac{\pi_\theta(y_\ell \mid x_{\text{DPO}})}{\pi_{\text{ref}}(y_\ell \mid x_{\text{DPO}})} \right). \quad (3)$$

For brevity, in the rest of the paper we will use $f_{\text{DPO}}(\theta)$ for $f_{\text{DPO}}(\theta; \mathcal{D}_{\text{DPO}}, \pi_{\text{ref}}, \beta)$, $g_{\text{DPO}}(\theta; x_{\text{DPO}}, y_w, y_\ell)$ for $g_{\text{DPO}}(\theta; x_{\text{DPO}}, y_w, y_\ell, \pi_{\text{ref}}, \beta)$, and $h_\beta(\theta; x_{\text{DPO}}, y_w, y_\ell)$ for $h_\beta(\theta; x, y_w, y_\ell, \pi_{\text{ref}})$. Note that $\frac{1}{N_1} \sum_{x_{\text{DPO}}, y_w, y_\ell \in \mathcal{D}_{\text{DPO}}} [g_{\text{DPO}}(\theta; x_{\text{DPO}}, y_w, y_\ell)] = \nabla f_{\text{DPO}}(\theta)$.

*SFT.* We denote the dataset used for SFT as $\mathcal{D}_{\text{SFT}} = \{x_{\text{SFT}}^{(i)}, y^{(i)}\}_{i=1}^{N_2}$, where $N_2$ is the number of data points. The SFT dataset consists of input $x^{(i)}$ and corresponding target outputs $y^{(i)}$ for all $i \in \{1, \ldots, N_2\}$. The objective used for fine-tuning $\theta$ for $\mathcal{D}_{\text{SFT}}$ can be given as

$$f_{\text{SFT}}(\theta; \mathcal{D}_{\text{SFT}}) := -\frac{1}{N_2} \sum_{x_{\text{SFT}}, y \in \mathcal{D}_{\text{SFT}}} \log(\pi_\theta(y \mid x)). \quad (4)$$

Given a data point $(x_{\text{SFT}}, y)$, the gradient estimate for the objective $f_{\text{SFT}}$ is given as

$$g_{\text{SFT}}(\theta; x_{\text{SFT}}, y) := -\nabla_\theta \pi_\theta(y \mid x_{\text{SFT}}) / \pi_\theta(y \mid x_{\text{SFT}}). \quad (5)$$

From this point, we will use $f_{\text{SFT}}(\theta)$ for $f_{\text{SFT}}(\theta; \mathcal{D}_{\text{SFT}})$. Note that $\frac{1}{N_2} \sum_{x_{\text{SFT}}, y \in \mathcal{D}_{\text{SFT}}} [g_{\text{SFT}}(\theta; x_{\text{SFT}}, y)] = \nabla f_{\text{SFT}}(\theta)$.

*Performance metric and trade-off.* In this work we investigate different methods for their performance on both DPO and SFT tasks, simultaneously. Thus, to evaluate the performance of a model $\theta$ on $f_{\text{DPO}}$ and $f_{\text{SFT}}$, we define the optimality gap of a mixture of objectives as

$$G_{\text{Mix}, \lambda}(\theta) := f_{\text{Mix}, \lambda}(\theta) - f_{\text{Mix}, \lambda}^*, \quad (6)$$

where $\lambda \in [0, 1]$, $f_{\text{Mix}, \lambda}(\theta) := \lambda f_{\text{DPO}}(\theta) + (1 - \lambda) f_{\text{SFT}}(\theta)$, and $f_{\text{Mix}, \lambda}^* = \min_{\theta \in \Theta} f_{\text{Mix}, \lambda}(\theta)$. Here $\lambda$ defines a trade-off between the DPO and SFT objective: a larger $\lambda$ results in more emphasis on the DPO performance compared to SFT. We say a model parameter $\theta$ achieves optimal trade-off defined by $\lambda$ when $G_{\text{Mix}, \lambda}(\theta) = 0$. We chose this metric because, as established in multi-objective optimization literature (Miettinen, 1999), the optimizer of $G_{\text{Mix}, \lambda}(\theta)$ for any $\lambda \in [0, 1]$ will be 'Pareto optimal'. This means that no other solution can optimize both objectives simultaneously, and the solution can be viewed as one of the optimal trade-off points for the problem of optimizing $f_{\text{DPO}}$ and $f_{\text{SFT}}$. Additionally, $G_{\text{Mix}, \lambda}(\theta)$ is differentiable when both $f_{\text{DPO}}$ and $f_{\text{SFT}}$ are differentiable, which facilitates the theoretical analysis of gradient-based methods.

## 3 WHY SEQUENTIAL DPO AND SFT IS SUBOPTIMAL?

This section studies the sequential DPO and SFT method commonly used in the continual training of aligned LLMs (see, e.g., Tang et al. (2020); Qi et al. (2023); Fang et al. (2024)). We give insights on why such a sequential training framework is suboptimal in terms of DPO and SFT trade-offs.

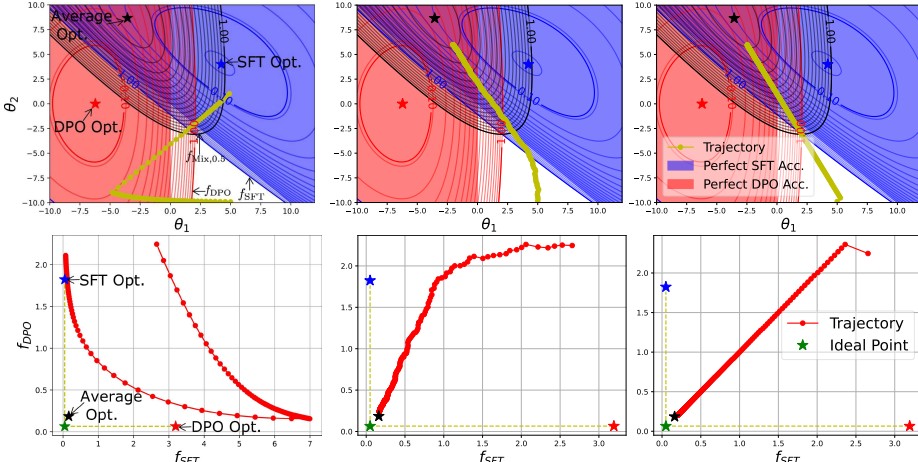

Figure 2: Consider a two-dimensional model and one data each for DPO and SFT optimization setting. **Sequential DPO and SFT (Left):** The model oscillates between the optima of DPO and SFT losses in weight space, resulting in a final trade-off that is significantly distant from the ideal point in loss space, where both DPO and SFT loss values are optimal. **ALRIGHT (Middle) / MAXRIGHT (Right):** The model directly navigates towards a point in weight space that is reasonably optimal for both DPO and SFT objectives (average optimum), achieving a final trade-off of DPO and SFT losses much closer to the ideal point compared to sequential DPO and SFT.

### 3.1 SEQUENTIAL TRAINING ALGORITHM AND ITS SUBOPTIMALITY

Following Rafailov et al. (2024), we first obtain a reference model $\pi_{\text{ref}}$ by performing SFT on the target outputs in the preference dataset $\mathcal{D}_{\text{DPO}}$. Given $\pi_{\text{ref}}$, we iteratively perform the DPO update as

$$\theta_{t+1}^1 = \Pi_\Theta \left( \theta_t^1 - \alpha_{1,t} g_{\text{DPO}}(\theta_t^1; x_{\text{DPO}}^t, y_w^t, y_\ell^t) \right), \quad \text{for } t = 1, 2, \ldots, T_{\text{DPO}} - 1 \qquad (7)$$

where $\alpha_{1,t}$ is the step size, $x_{\text{DPO}}^t, y_w^t, y_\ell^t \sim \mathcal{D}_{\text{DPO}}$, $g_{\text{DPO}}$ is defined in (2), and $T_{\text{DPO}}$ is the number of DPO iterations. Given the aligned model parameter $\theta_{T_{\text{DPO}}}^1$, we next perform SFT updates as follows:

$$\theta_{t+1}^2 = \Pi_\Theta \left( \theta_t^2 - \alpha_{2,t} g_{\text{SFT}}(\theta_t^2; x_{\text{SFT}}^t, y^t) \right), \quad \text{for } t = 1, 2, \ldots, T_{\text{SFT}} - 1 \qquad (8)$$

where $\theta_1^2 := \theta_{T_{\text{DPO}}}^1$, $x_{\text{SFT}}^t, y^t \sim \mathcal{D}_{\text{SFT}}$, $g_{\text{SFT}}$ is defined in (5), and $T_{\text{SFT}}$ is the number of SFT iterations. This process is summarized in Algorithm 1. We next study why the sequential training is suboptimal.

*A toy illustration of suboptimality.* At a given phase of Algorithm 1, the algorithm only focuses on optimizing one objective (either $f_{\text{DPO}}$ or $f_{\text{SFT}}$) and ignores the other. This results in the model oscillating between the optimums of two objectives, without converging to a point that is 'reasonably optimal' for both objectives $f_{\text{DPO}}$ and $f_{\text{SFT}}$. We first illustrate this on a toy example (see example details in Appendix D.1). The results are depicted in Figure 2. For the weight space trajectory (Figure 2 upper-left), although there is a region that is optimal for both DPO and SFT, the sequential DPO and SFT method fails to reach this region due to its focus on one objective at a given phase. Furthermore, from the loss space trajectory (Figure 2 lower-left), the model oscillates between

---

**Algorithm 1** Sequential DPO and SFT

1: Input $\mathcal{D}_{\text{DPO}}, \mathcal{D}_{\text{SFT}}, \{\alpha_{1,t}\}_{t=1}^{T_{\text{DPO}}}, \{\alpha_{2,t}\}_{t=1}^{T_{\text{SFT}}}$
2: **Phase 1: Optimize for $f_{\text{DPO}}$**
3: Initialize $\theta_1^1 := \theta_1 \in \Theta$
4: **for** $t = 1, \ldots, T_{\text{DPO}} - 1$ **do**
5:     Sample $x_{\text{DPO}}^t, y_w^t, y_\ell^t \sim \mathcal{D}_{\text{DPO}}$
6:     Update $\theta_{t+1}^1 = \Pi_\Theta \left( \theta_t^1 - \alpha_{1,t} g_{\text{DPO}}(\theta_t^1; x_{\text{DPO}}^t, y_w^t, y_\ell^t) \right)$
7: **end for**
8: Set $\hat{\theta}_{\text{DPO}} := \theta_{T_{\text{DPO}}}^1$
9: **Phase 2: Optimize for $f_{\text{SFT}}$**
10: Initialize $\theta_1^2 := \hat{\theta}_{\text{DPO}}$
11: **for** $t = 1, \ldots, T_{\text{SFT}} - 1$ **do**
12:     Sample $x_{\text{SFT}}^t, y^t \sim \mathcal{D}_{\text{SFT}}$
13:     Update $\theta_{t+1}^2 = \Pi_\Theta \left( \theta_t^2 - \alpha_{2,t} g_{\text{SFT}}(\theta_t^2; x_{\text{SFT}}^t, y^t) \right)$
14: **end for**
15: Output $\hat{\theta}_{\text{Seq}} := \theta_{T_{\text{SFT}}}^2$

---

extreme trade-offs for DPO and SFT, and ends up at a point far away from the ideal point.

## 3.2 THEORETICAL ANALYSIS OF SUBOPTIMALITY IN SEQUENTIAL METHOD

In this section, we provide a theoretical result on the suboptimal trade-offs between DPO and SFT in sequential training. We view the LLM as a policy $\pi_\theta$ that is characterized by a softmax:

$$\pi_\theta(y \mid x) := \frac{\exp(\theta^\top \phi_{y,x})}{\sum_{y' \in \mathcal{Y}} \exp(\theta^\top \phi_{y',x})},$$

where $\phi_{y,x}$ is a feature vector corresponding to the input $x$ and the target output $y$. Furthermore, reference policy $\pi_{\text{ref}}$ is similarly parameterized by a fixed parameter $\theta_{\text{ref}}$.

*Remark* 3.1. The softmax characterization is also used in previous theoretical works on RLHF; see, e.g., Zhu et al. (2023a). When the trainable parameter is the output projection weights, the LLM is fully characterized by the softmax. In other scenarios like performing LoRA Hu et al. (2021) on attention matrices or full-parameter training, we believe this characterization still provides valuable insights and our result will be verified empirically later in the experimental section.

We make the following mild assumption on the features.

**Assumption 3.1** (Bounded feature). For all $x \in \mathcal{X}$ and $y \in \mathcal{Y}$, there exists $\Phi > 0$ such that $\|\phi_{y,x}\| \leq \Phi$.

We can then have the following result for the sub-optimality of the output of Algorithm 1 to the optimum of some combination of functions $f_{\text{DPO}}$ and $f_{\text{SFT}}$ in terms of $G_{\text{Mix},\lambda}$.

**Theorem 3.1** (Lower bound for sequential method performance ). *Consider Algorithm 1 with $T_{DPO} = T_{SFT} = T$ under Assumption 3.1. Then there exists data $\mathcal{D}_{DPO}$ and $\mathcal{D}_{SFT}$ such that given any $\lambda \in (0, 1)$, Algorithm 1 with any sufficiently large $T$ has non-diminishing performance gap:*

$$\mathbb{E}\left[\lambda f_{DPO}(\hat{\theta}_{Seq}) + (1-\lambda) f_{SFT}(\hat{\theta}_{Seq}) - \min_{\theta \in \Theta} \left(\lambda f_{DPO}(\theta) + (1-\lambda) f_{SFT}(\theta)\right)\right] = \Omega(1), \tag{9}$$

*where the expectation $\mathbb{E}[\,\cdot\,]$ is taken over the randomness of Algorithm 1.*

The above result suggests that there exists DPO and SFT optimization problems such that given any trade-off between DPO and SFT defined by $\lambda \in (0, 1)$, the sequential method suffers from constant suboptimality gap, even when optimized for a large number of iterations. The reason for the constant suboptimality gap is the sequential method described in Algorithm 1 suffers from forgetting, and cannot appropriately optimize both the DPO and SFT objectives. In the next section, we explore alternatives to the sequential method that can resolve this issue.

## 4 PROPOSED ALTERNATING TRAINING METHODS

In this section we introduce new algorithms with theoretically convergence guarantees, which also outperforms sequential DPO and SFT empirically.

### 4.1 ALRIGHT FOR JOINT DPO AND SFT

The main disadvantage of using Algorithm 1 for DPO and SFT optimization is that at a given phase of the algorithm, the model is updated with respective to only one objective. In contrast, it is computationally intensive, if not prohibitive, to optimize a linear combination of both DPO and SFT objectives. This is because, although the objectives share a single parameter, constructing two computational graphs (one per objective) in standard machine learning libraries requires significant additional memory, particularly for LLMs. To alleviate these problems, we propose to alternate between optimizing for DPO and SFT objectives, based on a given preference for each objective. For this purpose, we can define a modified objective

$$f_{\text{Alt},\lambda}(\theta; i) = \mathbb{I}_{i=1} f_{\text{DPO}}(\theta) + \mathbb{I}_{i=0} f_{\text{SFT}}(\theta), \tag{10}$$

where $i \sim \text{Bern}(\lambda)$, $\text{Bern}(\lambda)$ is the Bernoulli distribution parameterized by $\lambda \in [0, 1]$, and $\mathbb{I}_A$ is the indicator function of event $A$. Hence, the objective in (10) behaves as $f_{\text{Mix},\lambda}$ in expectation, i.e.

$$\mathbb{E}_{i \sim \text{Bern}(\lambda)} \left[f_{\text{Alt},\lambda}(\theta; i)\right] = \lambda f_{\text{DPO}}(\theta) + (1 - \lambda) f_{\text{SFT}}(\theta) = f_{\text{Mix},\lambda}(\theta). \tag{11}$$

| **Algorithm 2** ALRIGHT | **Algorithm 3** MAXRIGHT |
|---|---|
| 1: Input $\mathcal{D}_{\text{DPO}}, \mathcal{D}_{\text{SFT}}, \{\alpha_t\}_{t=1}^T, \lambda \in [0,1]$ | 1: Input $\mathcal{D}_{\text{DPO}}, \mathcal{D}_{\text{SFT}}, \{\alpha_t\}, \lambda \in [0,1]$. Initialize $\theta_1 \in \Theta$ |
| 2: Initialize $\theta_1 \in \Theta$ | 2: **for** $t = 1, \ldots, T-1$ **do** |
| 3: **for** $t = 1, \ldots, T-1$ **do** | 3: $\quad$ Sample $x_{\text{DPO}}^t, y_w^t, y_\ell^t \sim \mathcal{D}_{\text{DPO}}$ and $x_{\text{SFT}}^t, y^t \sim \mathcal{D}_{\text{SFT}}$ |
| 4: $\quad$ Sample $i_t \sim \text{Bern}(\lambda)$ | 4: $\quad$ Evaluate |
| 5: $\quad$ **if** $i_t = 1$ **then** | $\quad\quad \bar{f}_{1,\lambda}(\theta_t) := \lambda\left(f_{\text{DPO}}(\theta_t; x_{\text{DPO}}^t, y_w^t, y_\ell^t) - f_{\text{DPO}}^*\right)$ |
| 6: $\quad\quad$ Sample $x_{\text{DPO}}^t, y_w^t, y_\ell^t \sim \mathcal{D}_{\text{DPO}}$ | $\quad\quad \bar{f}_{2,\lambda}(\theta_t) := (1-\lambda)\left(f_{\text{SFT}}(\theta_t; x_{\text{SFT}}^t, y^t) - f_{\text{SFT}}^*\right)$ |
| 7: $\quad\quad \theta_{t+1} = \Pi_\Theta\left(\theta_t - \alpha_t g_{\text{DPO}}(\theta_t; x_{\text{DPO}}^t, y_w^t, y_\ell^t)\right)$ | 5: $\quad$ **if** $\bar{f}_{1,\lambda}(\theta_t) \geq \bar{f}_{2,\lambda}(\theta_t)$ **then** |
| 8: $\quad$ **else** | 6: $\quad\quad \theta_{t+1} = \Pi_\Theta\left(\theta_t - \alpha_t g_{\text{DPO}}(\theta_t; x_{\text{DPO}}^t, y_w^t, y_\ell^t)\right)$ |
| 9: $\quad\quad$ Sample $x_{\text{SFT}}^t, y^t \sim \mathcal{D}_{\text{SFT}}$ | 7: $\quad$ **else** |
| 10: $\quad\quad \theta_{t+1} = \Pi_\Theta\left(\theta_t - \alpha_t g_{\text{SFT}}(\theta_t; x_{\text{SFT}}^t, y^t)\right)$ | 8: $\quad\quad \theta_{t+1} = \Pi_\Theta\left(\theta_t - \alpha_t g_{\text{SFT}}(\theta_t; x_{\text{SFT}}^t, y^t)\right)$ |
| 11: $\quad$ **end if** | 9: $\quad$ **end if** |
| 12: **end for** | 10: **end for** |
| 13: Output $\hat{\theta}_{\text{AL}} := \theta_T$ | 11: Output $\hat{\theta}_{\text{MAX}} := \theta_T$ |

For optimizing $\mathbb{E}_{i \sim \text{Bern}(\lambda)}\left[f_{\text{Alt},\lambda}(\theta; i)\right]$, we first sample $i_t \sim \text{Bern}(\lambda)$ per iteration, which determines the objective to be updated. Specifically, if $i_t = 1$, we update $\theta$ with respect to DPO objective as

$$\theta_{t+1} = \Pi_\Theta\left(\theta_t - \alpha_t g_{\text{DPO}}(\theta_t; x_{\text{DPO}}^t, y_w^t, y_\ell^t)\right), \tag{12}$$

where $x_{\text{DPO}}^t, y_w^t, y_\ell^t \sim \mathcal{D}_{\text{DPO}}$, and $\alpha_t$ is the learning rate. If $i_t = 0$, $\theta$ is updated using SFT objective as

$$\theta_{t+1} = \Pi_\Theta\left(\theta_t - \alpha_t g_{\text{SFT}}(\theta_t; x_{\text{SFT}}^t, y^t)\right), \tag{13}$$

where $x_{\text{SFT}}^t, y^t \sim \mathcal{D}_{\text{SFT}}$. This process is summarized in Algorithm 2. Unlike the sequential method that focuses on optimizing one objective at a time, the ALRIGHT approach integrates both objectives simultaneously, allowing the model to balance alignment and fine-tuning performance. In Figure 2 (Middle), we can see how this alternating navigates the model to a point where the trade-off between DPO and SFT is significantly better than the sequential approach.

Next, we provide the convergence guarantee of Algorithm 2 to a given DPO-SFT trade-off.

**Theorem 4.1** (Upper bound for alternating method performance). *Consider Algorithm 2 with* $\alpha_t = \alpha_0/\sqrt{T}$ *for all* $t \in \{1, \ldots, T\}$ *and* $\alpha_0 > 0$. *Then, under Assumption 3.1, for any* $\lambda \in [0,1]$, *we have*

$$\mathbb{E}\left[\lambda f_{DPO}(\hat{\theta}_{AL}) + (1-\lambda) f_{SFT}(\hat{\theta}_{AL}) - \min_{\theta \in \Theta}\left(\lambda f_{DPO}(\theta) + (1-\lambda) f_{SFT}(\theta)\right)\right] = \mathcal{O}\left(\frac{\log T}{\sqrt{T}}\right). \tag{14}$$

*Remark* 4.1. The above result implies that the performance metric diminishes with increasing $T$, thus we can achieve an arbitrary trade-off between DPO and SFT defined by $\lambda$ up to arbitrary optimality, by increasing the number of iterations $T$. This is in contrast to the lower bound result in Theorem 3.1 established for sequential training: there exist data sets such that the sequential method never approaches optimal trade-off, even when trained for larger number of iterations.

While ALRIGHT offers theoretical convergence guarantees for any arbitrary trade-off in expectation, the alternation between optimizing DPO and SFT objectives occurs randomly based on a predetermined probability, which may introduce additional noise in the updates. This raises the natural question: Can we design a performance-aware, *adaptive alternating* optimization method with minimal additional computational resource usage compared to ALRIGHT? In the next section, we will propose an alternative post-training method that adaptively selects the objective to optimize.

### 4.2 MAXRIGHT FOR JOINT DPO AND SFT

In this section, we introduce a method that can adaptively choose objective to be optimized based on the current performance of $\theta$, which can lead to faster convergence to a point that can perform well for both DPO and SFT objectives. To this end, we first compare the current model's performance on $f_{\text{DPO}}$ and $f_{\text{SFT}}$. Define the maximum (weighted) sub-optimality gap as

$$f_{\text{Max},\lambda}(\theta) = \max\left(\lambda(f_{\text{DPO}}(\theta) - f_{\text{DPO}}^*), (1-\lambda)(f_{\text{SFT}}(\theta) - f_{\text{SFT}}^*)\right), \tag{15}$$

where $f_{\text{DPO}}^* = \min_{\theta \in \Theta} f_{\text{DPO}}(\theta)$ (similarly for $f_{\text{SFT}}(\theta)$), and $\lambda \in [0,1]$. The idea is to optimize this maximum sub-optimality to reach a balance between the two $\lambda$-scaled objectives. Define the index of

the objective with maximum (weighted) sub-optimality gap as $i_t = \arg\max_i \bar{f}_i(\theta_t)$, where

$$\bar{f}_{1,\lambda}(\theta_t) \coloneqq \lambda \left( f_{\text{DPO}}(\theta_t; x_{\text{DPO}}^t, y_w^t, y_\ell^t) - f_{\text{DPO}}^* \right), \quad \text{and} \tag{16}$$

$$\bar{f}_{2,\lambda}(\theta_t) \coloneqq (1 - \lambda) \left( f_{\text{SFT}}(\theta_t; x_{\text{SFT}}^t, y^t) - f_{\text{SFT}}^* \right), \tag{17}$$

where $x_{\text{DPO}}^t, y_w^t, y_\ell^t \sim \mathcal{D}_{\text{DPO}}$ and $x_{\text{SFT}}^t, y^t \sim \mathcal{D}_{\text{SFT}}$. Accordingly, we can update $\theta$ with respect to DPO objective using update (12) when $i_t = 1$ (or equivalently, when $\bar{f}_{1,\lambda}(\theta_t) \geq \bar{f}_{2,\lambda}(\theta_t)$), and update $\theta$ with respect to DPO objective using update (13) otherwise. This process is summarized in Algorithm 3. We can see in the toy illustration (Figure 2 Right), that MAXRIGHT can converge closer to the ideal point more directly compared to ALRIGHT, due to its performance based update of objectives.

*Remark* 4.2. It is a well-known fact in multi-objective optimization literature (Miettinen, 1999) that under some assumptions on the problem setup, the solution of problem (15) for any $\lambda \in [0, 1]$ is guaranteed to be Pareto optimal (i.e. no other solution can further optimize both the objectives simultaneously). Furthermore, unlike the ALRIGHT algorithm, MAXRIGHT requires prior knowledge or computation of $f_{\text{DPO}}^*$ and $f_{\text{SFT}}^*$, adding to its overall computational budget. However, this computation is performed once and can be reused across different implementations with varying $\lambda$. Details on approximating $f_{\text{DPO}}^*$ and $f_{\text{SFT}}^*$ are provided in Appendix D.

Even though MAXRIGHT allows one to compute the index needed for selecting the objective with a maximum (weighted) sub-optimality gap, in practice evaluating both objectives can be memory intensive, and only one objective is updated at a given iteration. To alleviate this issue, we propose to do simultaneous evaluations only every $k$ steps. We call a time step that simultaneous evaluation is done as a 'max evaluation step'. At such time step $t = t_0$, we compute $i_{t_0}$, and update the corresponding objective as in Algorithm 3. After the update, we store the computed (weighted) sub-optimality gap as $\bar{f}_{1,\lambda}^{\text{stale},t_0} = \bar{f}_{1,\lambda}(\theta_{t_0})$ and $\bar{f}_{2,\lambda}^{\text{stale},t_0} = \bar{f}_{2,\lambda}(\theta_{t_0})$. Then, for every iteration before the next max evaluation step $t_0 + k$, we choose the index of the objective to be optimized as

$$i_{t_0+k'} = \arg\max_i \bar{f}_{i,\lambda}^{\text{stale},t_0}, \tag{18}$$

where $k' < k$. Once the index is computed, we update the corresponding objective following (12) or (13), and update the stale (weighted) sub-optimality gap as

$$\bar{f}_{i,\lambda}^{\text{stale},t_0} = \bar{f}_{i,\lambda}(\theta_{t_0+k'}), \quad \text{if } i_{t_0+k'} = i, \tag{19}$$

where $i \in \{1, 2\}$. This process is summarized in Appendix B. With this modification, we can match the evaluation and gradient computation complexity of Algorithm 2 in most iterations, at the expense of degraded accuracy in choosing $i_t$.

## 5 RELATED WORK

**RLHF.** The most fundamental form of RLHF was introduced by Christiano et al. (2017) and has been successfully used for aligning LLMs in many works such as OpenAI (2022); Ouyang et al. (2022); Bai et al. (2022a;b); Sun et al. (2024). There have been many works on RLHF for LLM alignment, including the more efficient direct preference optimization (Rafailov et al., 2024; Xu et al., 2024; Lee et al., 2024; Zhong et al., 2024), reference model free preference alignment (Meng et al., 2024; Hong et al., 2024b), generalized RLHF (Azar et al., 2024; Munos et al., 2023), safe RLHF (Dai et al., 2023), group preference learning (Zhao et al., 2023; Chakraborty et al., 2024), and theory or understanding of RLHF (Zhu et al., 2023a; Shen et al., 2024; Xiong et al., 2024; Wang et al., 2023; Kirk et al., 2023). In this work, we consider DPO (Rafailov et al., 2024) which has been used in training many popular open-source LLMs (Abdin et al., 2024; Dubey et al., 2024).

**SFT.** Another important step before using a pre-trained LLM in downstream applications is SFT (Howard & Ruder, 2018; Devlin, 2018; Wei et al., 2021; Zhu et al., 2023b; Zhang et al., 2023b). In recent years, there have been a large body of work on efficient LLM SFT; see, e.g., zeroth-order fine-tuning (Malladi et al., 2023a), quantized fine-tuning (Kim et al., 2024; Li et al., 2023b), parameter-efficient fine-tuning (Chen et al., 2023a; Zhang et al., 2023a; Shi & Lipani, 2023; Chen et al., 2023b; Nikdan et al., 2024), truthful fine-tuning (Tian et al., 2023), robust fine-tuning (Tian et al., 2024), SFT with data selection (Lu et al., 2023; Kang et al., 2024; Zhao et al., 2024), self-play fine-tuning (Chen et al., 2024) and understanding of LLM fine-tuning (Malladi et al., 2023b).

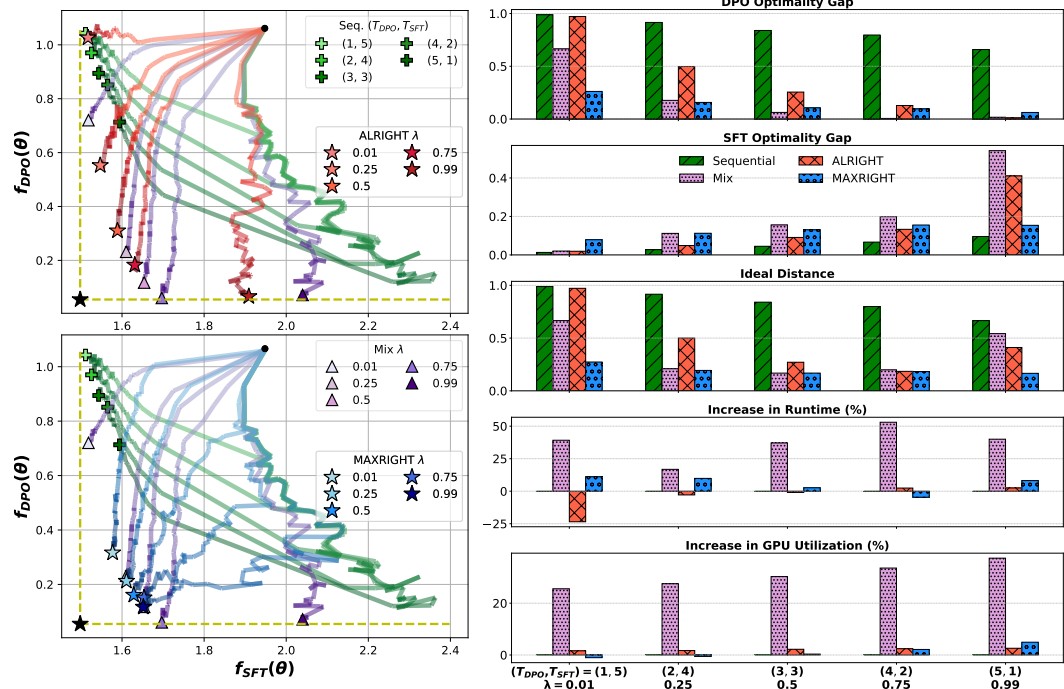

Figure 3: Comparison of proposed methods with first DPO then SFT using PYTHIA-1B model. **Left:** Training trajectories in the objective space (e.g., DPO and SFT objectives). **Right:** Performance comparison across multiple evaluation metrics, including optimality gap for DPO and SFT objectives, ideal distance, runtime, and GPU utilization. The bar charts highlight the trade-offs and resource efficiency of each method for different choices of $(T_{\text{DPO}}, T_{\text{SFT}})$ or $\lambda$.

**Sequential RLHF and SFT Issues.** Given the importance of both RLHF and SFT in LLM post-training, they are implemented as a sequential recipe (one after the other) on a pre-trained LLM. However, recent studies show that this either hinders the alignment performance (Qi et al., 2023), or fine-tuning performance (Ouyang et al., 2022), depending on the order of applying RLHF and SFT.

**Reconciling RLHF and SFT.** Several methods have been proposed to reconcile RLHF and SFT in post training. One line of work attempt to remove a separate RLHF phase by adding regularization to SFT objective (Hong et al., 2024a), reformulation of SFT objective (Hua et al., 2024), and joint training using demonstrations and RLHF (Li et al., 2024). However, these methods restrict the use of different datasets for RLHF and SFT. Adaptive model averaging (AMA) (Lin et al., 2023) optimizes RLHF and SFT separately to balance objectives, but the original AMA is computationally expensive, requiring three sets of model parameters. While a memory-efficient AMA variant exists, its reliance on imitation learning weakens ties to the original objectives, and the relationship between AMA weights and optimal trade-offs remains unclear.

# 6 EXPERIMENTS

In this section we compare the proposed methods with some existing baselines, in terms of their Pareto-front performance, and resource consumption such as computation time and memory usage.

## 6.1 EXPERIMENTAL SETUP

In this section, we introduce models, datasets, baselines, and evaluation metrics used to evaluate the performance of our proposed methods. Additional experiment details are provided in Appendix D.

**Models.** We employ two architectures. The first is ELEUTHERAI/PYTHIA-1B[1], a widely used model balancing computational efficiency and performance. Despite not being designed for downstream tasks, it matches or exceeds models like OPT and GPT-Neo of similar size. We use it to assess optimization dynamics, performance trade-offs, and resource usage of proposed methods. The second

---

[1] https://huggingface.co/EleutherAI/pythia-1b

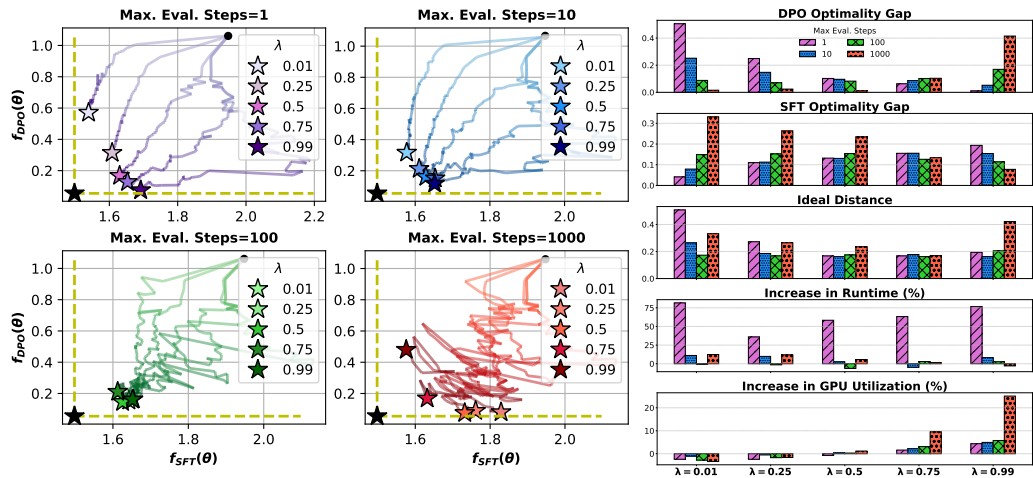

Figure 4: Comparison of different choices of evaluation steps for MAXRIGHT with Pythia-1b model.

is META-LLAMA/META-LLAMA-3-8B[2], a larger model suited for fine-tuning and downstream real-world tasks. Both models are fine-tuned using Low-Rank Adaptation (LoRA) (Hu et al., 2021).

**Datasets.** For the DPO dataset, we use the DAHOAS/RM-HH-RLHF dataset, which contains human feedback data designed to align models to human preference. For the SFT phase, we use the VICGALLE/ALPACA-GPT4 dataset, which consists of English instruction-following data generated by GPT-4 using Alpaca prompts, designed for fine-tuning LLMs.

**Baseline Methods.** Comparing the performance of ALRIGHT and MAXRIGHT, we use the following baselines: Mix of DPO and SFT ('Mix'), which simultaneously optimizes both DPO and SFT objectives by optimizing a convex combination of the objectives, and Sequential DPO and SFT ('Sequential'), where DPO and SFT objectives are optimized one after the other.

**Evaluation Metrics.** To assess the performance of each method with respect to the DPO and SFT objectives, we utilize several evaluation metrics. For evaluating the DPO objective, we measure the **optimality gap** as $f_{\text{DPO}}(\theta) - f_{\text{DPO}}^*$, where $f_{\text{DPO}}^*$ is approximated by independently optimizing the DPO objective for the same number of iterations as used for the baselines and proposed methods. The optimality gap for the SFT objective is similarly defined using the optimal value $f_{\text{SFT}}^*$, obtained by separately optimizing the SFT objective. To evaluate overall performance, we use the **ideal distance** metric, which represents the Euclidean distance between the final iterate produced by the method and the point corresponding to optimal values for both DPO and SFT objectives: $\sqrt{(f_{\text{DPO}}(\theta) - f_{\text{DPO}}^*)^2 + (f_{\text{SFT}}(\theta) - f_{\text{SFT}}^*)^2}$. For resource efficiency, we compare the **percentage increase in runtime** relative to the corresponding sequential implementation, e.g., the percentage runtime increase of Alternating DPO and SFT with $\lambda = 0.01$ compared to Sequential DPO and SFT with $(T_{\text{DPO}}, T_{\text{SFT}}) = (1, 5)$. Additionally, we compute the **percentage increase in GPU utilization** for each method relative to the baselines. Further details on these metrics are provided in Appendix D due to space constraints. Furthermore, for evaluating the real-world performance of proposed methods compared to baselines, we use the following benchmarks: **MMLU**(Hendrycks et al., 2020), a benchmark with multiple-choice questions across 57 diverse tasks; **Win rate**, which is calculated as the proportion of times a model's response is preferred by an evaluator over a baseline in head-to-head comparisons. For this purpose, we use ALPACAEVAL(Li et al., 2023a) framework with GPT-4-TURBO as the evaluator and DAHOAS/RM-HH-RLHF test data as the baseline.

## 6.2 EXPERIMENT RESULTS

In this section we illustrate and discuss the empirical results obtained under the experiment setup introduced in the previous section.

**ALRIGHT provides better control over the trade-off compared to Sequential.** As shown in the top left plot of Figure 3, the optimization trajectories for DPO followed by SFT illustrate that the set of final models produced by ALRIGHT, for various values of $\lambda$, is more evenly distributed in the objective space. This distribution forms a Pareto front, indicating that no model is strictly worse than

---

[2]https://huggingface.co/meta-llama/Meta-Llama-3-8B

| $\lambda/(T_{\text{SFT}}, T_{\text{DPO}})$ | MMLU (1-shot) (%) | | | Win rate (%) | | |
|---|---|---|---|---|---|---|
| | $0.25/(3,1)$ | $0.5/(2,2)$ | $0.75/(1,3)$ | $0.25/(3,1)$ | $0.5/(2,2)$ | $0.75/(1,3)$ |
| Sequential | 73.18 | 72.80 | 72.68 | 57.19 | 65.62 | 59.38 |
| Mix | 73.45 | 73.40 | 72.29 | 81.88 | 84.22 | **88.42** |
| ALRIGHT | **74.66** | 72.65 | **75.50** | **88.28** | 85.78 | 87.34 |
| MAXRIGHT | 72.35 | **73.42** | 74.24 | 86.56 | **86.09** | 83.75 |

Table 1: Comparison of Win rate and MMLU (1-shot) for different methods using LLAMA3-8B.

another with respect to both objectives. Moreover, the spread of these models is comparable to that of the Mix method. In contrast, Sequential tends to produce models that are biased towards the SFT objective, even when $T_{\text{DPO}}$ is significantly larger than $T_{\text{SFT}}$ (e.g., $(T_{\text{DPO}}, T_{\text{SFT}}) = (5, 1)$).

**MAXRIGHT achieves near-ideal performance compared to other methods.** As illustrated in the top left plot of Figure 3, the optimization trajectories for DPO followed by SFT show that the set of final models produced by MAXRIGHT, for different values of $\lambda$, converge closer to the ideal point compared to other methods. This behavior is further supported by the Ideal Distance comparison in the right plot of Figure 3, where MAXRIGHT consistently achieves the best ideal distance performance across all $\lambda$ values. We attribute this advantage to the adaptive nature of MAXRIGHT, which dynamically selects the objective to update based on performance, rather than adhering to a fixed schedule like ALRIGHT. This adaptability is particularly beneficial in heavily over-parameterized settings, where models have the capacity to approach ideal performance.

**ALRIGHT and MAXRIGHT require minimal additional resources compared to Sequential and significantly lower than Mix.** As shown in Figure 3 (right, Increase in Runtime (%) and Increase in GPU Utilization (%)), the additional computational resources required by different implementations of ALRIGHT and MAXRIGHT are minimal (or even negative) relative to their Sequential counterparts. In contrast, Mix incurs substantial additional resource usage, with increases of over 50% in runtime and more than 35% in GPU utilization, despite achieving similar performance metrics to ALRIGHT and MAXRIGHT.

**Effect of maximum evaluation step for memory-efficient MAXRIGHT.** Figure 4 illustrates the influence of maximum evaluation step choices in memory efficient MAXRIGHT on optimization trajectories and resource usage. For low values (e.g., 1), the algorithm closely follows the trade-off determined by $\lambda$, keeping the solutions concentrated near the ideal point (e.g., compared to ALRIGHT), but incurs high runtime due to frequent evaluations. In contrast, high values (e.g., 1000) cause significant oscillations in the objective space, failing to maintain the desired trade-off and resulting in increased GPU utilization from excessive SFT updates. The aforementioned oscillation leads to poor ideal distance performance as the model drifts away from the ideal point.

**ALRIGHT and MAXRIGHT significantly outperform Sequential on real-world tasks with minimal additional resources.** Table 1 presents a performance comparison of Sequential, Mix, ALRIGHT, and MAXRIGHT using the MMLU benchmark and win rate. ALRIGHT and MAXRIGHT consistently surpass the baselines on the MMLU benchmark and achieve a significantly higher win rate than Sequential across all $\lambda$ values considered. On both evaluation metrics, either ALRIGHT or MAXRIGHT performs on par with or better than Mix. Furthermore, Figure 7 in Appendix D shows that ALRIGHT and MAXRIGHT require minimal additional resources, while Mix incurs significantly higher resource usage (up to 58% increase in runtime and 7% increase in GPU utilization).

## 7 CONCLUSIONS AND DISCUSSION

In this paper, we have shown both theoretically and empirically that the widely adopted sequential approach to post-training LLMs with RLHF and SFT is sub-optimal, as the model gradually forgets the effects of the initial stage during the second stage of training. Our proposed ALRIGHT and MAXRIGHT methods address this issue, with ALRIGHT providing theoretical convergence guarantees and MAXRIGHT demonstrating strong empirical performance. Notably, this improvement is achieved with minimal additional computational cost, making these methods practical and efficient alternatives for enhancing both the performance and preference alignment of LLMs.

## 8 REPRODUCIBILITY STATEMENT

For the theoretical results given in the main text, we provide proofs in Appendix A. For all the experiment results provided in the main text and in the Appendix, we provide the implementation details in Appendix D. We will provide the code for implementing the main experiments in the discussion forums once they are open when the paper is under review, and the code will be released in GitHub upon acceptance of the paper.

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

**Supplementary Material for "Mitigating Forgetting in LLM Supervised Fine-Tuning and Preference Learning"**

# Table of Contents

# A  PROOFS FOR THEORETICAL RESULTS

In this section, we provide the proofs for the main theoretical results of the paper, Theorems 3.1 and 4.1. The proof is organized as follows: In Appendix A.1, we establish some fundamental claims regarding the problem setup, such as the convexity of the objectives and the boundedness of the second moment of the stochastic gradients. Then, in Appendix A.2, we prove the lower bound stated in Theorem 3.1 along with several supporting lemmas. Finally, in Appendix A.3, we prove the upper bound stated in Theorem 4.1, accompanied by additional supporting lemmas.

For conciseness, in this section we omit the subscripts DPO and SFT of the input $x$, but whether $x$ belongs to $\mathcal{D}_{\text{DPO}}$ or $\mathcal{D}_{\text{SFT}}$ can be inferred from the context. Furthermore, for brevity, we denote the averaging over the datasets $\frac{1}{N_1} \sum_{x,y_w,y_\ell \in \mathcal{D}_{\text{DPO}}}$ and $\frac{1}{N_2} \sum_{x,y \in \mathcal{D}_{\text{SFT}}}$ by $\mathbb{E}_{x,y_w,y_\ell \sim \mathcal{D}_{\text{DPO}}}$ and $\mathbb{E}_{x,y \sim \mathcal{D}_{\text{SFT}}}$, respectively.

## A.1  BASIC CLAIMS ON THE PROBLEM SETUP

Before going to the proofs of the main results, we can have the following basic claims on the problem setup.

**Proposition A.1** (Bounded second moment of stochastic gradient)**.** *For all $\theta \in \Theta$, there exist $M_1, M_2 > 0$ such that*

$$\mathbb{E}_{x,y_w,y_\ell \sim \mathcal{D}_{DPO}}[\|g_{DPO}(\theta; x, y_w, y_\ell)\|^2] \leq M_1^2, \tag{20}$$

$$\mathbb{E}_{x,y \sim \mathcal{D}_{SFT}}[\|g_{SFT}(\theta; x, y)\|^2] \leq M_2^2. \tag{21}$$

*Proof.* Considering $g_{\text{DPO}}$, we can first simplify $h_\beta$ defined in (3) under softmax parameterization of $\pi_\theta$ and $\pi_{\text{ref}}$ as

$$h_\beta(\theta; x, y_w, y_\ell) = \beta \log\left(\frac{\pi_\theta(y_w \mid x)}{\pi_{\text{ref}}(y_w \mid x)}\right) - \beta \log\left(\frac{\pi_\theta(y_\ell \mid x)}{\pi_{\text{ref}}(y_\ell \mid x)}\right)$$

$$= \beta(\theta - \theta_{\text{ref}})^\top (\phi_{y_w,x} - \phi_{y_\ell,x}). \tag{22}$$

Then we can simplify $g_{\mathrm{DPO}}$ as

$$g_{\mathrm{DPO}}(\theta; x, y_w, y_\ell) = -\left(1 - \sigma(h_\beta(\theta; x, y_w, y_\ell, \pi_{\mathrm{ref}}))\right) \nabla_\theta h_\beta(\theta; x, y_w, y_\ell, \pi_{\mathrm{ref}})$$
$$= -\beta \left(1 - \sigma\left(\beta(\theta - \theta_{\mathrm{ref}})^\top (\phi_{y_w,x} - \phi_{y_\ell,x})\right)\right) (\phi_{y_w,x} - \phi_{y_\ell,x}). \quad (23)$$

We can then bound the norm of $g_{\mathrm{DPO}}$ as

$$\|g_{\mathrm{DPO}}(\theta; x, y_w, y_\ell)\|^2 = \beta^2 \left(1 - \sigma\left(\beta(\theta - \theta_{\mathrm{ref}})^\top (\phi_{y_w,x} - \phi_{y_\ell,x})\right)\right)^2 \|\phi_{y_w,x} - \phi_{y_\ell,x}\|^2$$
$$\leq \beta^2 \|\phi_{y_w,x} - \phi_{y_\ell,x}\|^2$$
$$\leq 4\beta^2 \Phi^2 =: M_1, \quad (24)$$

where the first inequality is due to the fact that $0 \leq \sigma(z) \leq 1$ for all $z \in \mathbb{R}$, second inequality is due to Cauchy-Schwarz inequality and Assumption 3.1. Taking expectation (average) over the dataset $\mathcal{D}_{\mathrm{DPO}}$ in (24) proves the first part of Proposition A.1. For proving the second part of the proposition, we start by simplifying the gradient of $\pi_\theta$ under softmax-parameterization, given by

$$\nabla_\theta \pi_\theta(y \mid x) = \nabla_\theta \frac{\exp(\theta^\top \phi_{y,x})}{\sum_{y' \in \mathcal{Y}} \exp(\theta^\top \phi_{y',x})}$$
$$= \frac{\exp(\theta^\top \phi_{y,x}) \sum_{y' \in \mathcal{Y}} \exp(\theta^\top \phi_{y',x}) \phi_{y,x} - \exp(\theta^\top \phi_{y,x}) \sum_{y' \in \mathcal{Y}} \exp(\theta^\top \phi_{y',x}) \phi_{y',x}}{\left(\sum_{y' \in \mathcal{Y}} \exp(\theta^\top \phi_{y',x})\right)^2}$$
$$= \left(\phi_{y,x} - \bar{\phi}_x(\theta)\right) \pi_\theta(y \mid x), \quad (25)$$

where

$$\bar{\phi}_x(\theta) := \frac{\sum_{y' \in \mathcal{Y}} \phi_{y',x} \exp(\theta^\top \phi_{y',x})}{\sum_{y' \in \mathcal{Y}} \exp(\theta^\top \phi_{y',x})}. \quad (26)$$

Then we can simplify $g_{\mathrm{SFT}}$ as

$$g_{\mathrm{SFT}}(\theta; x, y) = -\frac{\nabla_\theta \pi_\theta(y \mid x)}{\pi_\theta(y \mid x)}$$
$$= -\left(\phi_{y,x} - \bar{\phi}_x(\theta)\right). \quad (27)$$

We can then bound the norm of $g_{\mathrm{SFT}}$ as

$$\|g_{\mathrm{SFT}}(\theta; x, y)\|^2 = \|\phi_{y,x} - \bar{\phi}_x(\theta)\|^2$$
$$\leq 4\Phi^2 =: M_2, \quad (28)$$

where the inequality is due to the Cauchy-Schwarz inequality and Jensen's inequality. Taking expectation (average) over the dataset $\mathcal{D}_{\mathrm{SFT}}$ in (28) proves the second part of Proposition A.1. $\square$

**Proposition A.2** (Convexity of objectives). *Under Assumption 3.1, the objectives $f_{DPO}$ and $f_{SFT}$ (defined in (1) and (4), respectively) are convex.*

*Proof.* The goal of the proof is to show the Hessians of the objectives $f_{\mathrm{DPO}}$ and $f_{\mathrm{SFT}}$ are semi-positive definite, under Assumption 3.1 and LLMs (both trainable and reference) modeled using softmax parameterization. First, considering $f_{\mathrm{DPO}}$, we can have

$$\nabla_\theta g_{\mathrm{DPO}}(\theta; x, y_w, y_\ell) = -\nabla_\theta \beta \left(1 - \sigma\left(\beta(\theta - \theta_{\mathrm{ref}})^\top (\phi_{y_w,x} - \phi_{y_\ell,x})\right)\right) (\phi_{y_w,x} - \phi_{y_\ell,x})$$
$$= \beta^2 \sigma\left(h_\beta(\theta; x, y_w, y_\ell)\right) (\phi_{y_w,x} - \phi_{y_\ell,x}) (\phi_{y_w,x} - \phi_{y_\ell,x})^\top \succeq 0, \quad (29)$$

where first equality is due to (23), and the semi-positive definiteness is due to the fact that $\beta > 0$, $0 \leq \sigma(z) \leq 1$ for all $z \in \mathbb{R}$, and $(\phi_{y_w,x} - \phi_{y_\ell,x}) (\phi_{y_w,x} - \phi_{y_\ell,x})^\top \succeq 0$ for all $\phi_{y_w,x}, \phi_{y_\ell,x}$. The convexity of $f_{\mathrm{DPO}}$ follows from the fact that

$$\nabla^2 f_{\mathrm{DPO}}(\theta) = \mathbb{E}_{x, y_w, y_\ell \sim \mathcal{D}_{\mathrm{DPO}}} \left[\nabla_\theta g_{\mathrm{DPO}}(\theta; x, y_w, y_\ell)\right]. \quad (30)$$

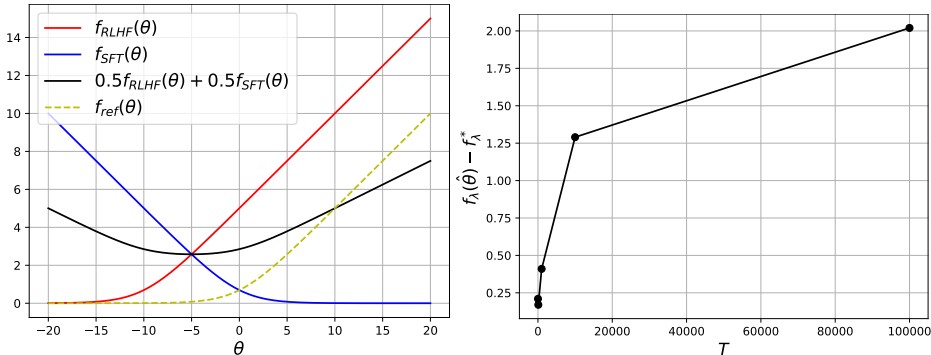

Figure 5: An Illustration of the example used for lower bound derivation in Theorem 3.1

Similarly, we can compute $\nabla_\theta g_{\mathrm{SFT}}$ as

$$\nabla_\theta g_{\mathrm{SFT}}(\theta; x, y) = -\nabla_\theta(\phi_{y,x} - \bar{\phi}_x(\theta))$$

$$= \nabla_\theta \frac{\sum_{y' \in \mathcal{Y}} \phi_{y',x} \exp(\theta^\top \phi_{y',x})}{\sum_{y' \in \mathcal{Y}} \exp(\theta^\top \phi_{y',x})}$$

$$= \sum_{y' \in \mathcal{Y}} \frac{\phi_{y',x} \phi_{y',x}^\top \exp(\theta^\top \phi_{y',x})}{\sum_{y' \in \mathcal{Y}} \exp(\theta^\top \phi_{y',x})}$$

$$- \left( \frac{\sum_{y' \in \mathcal{Y}} \phi_{y',x} \exp(\theta^\top \phi_{y',x})}{\sum_{y' \in \mathcal{Y}} \exp(\theta^\top \phi_{y',x})} \right) \left( \frac{\sum_{y' \in \mathcal{Y}} \phi_{y',x} \exp(\theta^\top \phi_{y',x})}{\sum_{y' \in \mathcal{Y}} \exp(\theta^\top \phi_{y',x})} \right)^\top, \quad (31)$$

where the first equality is due to (27). To establish the semi-positivedefiniteness of $\nabla_\theta g_{\mathrm{SFT}}$, cosider any $v$ with same dimension as $\theta$, and let $p_{y,x} = \frac{\exp(\theta^\top \phi_{y,x})}{\sum_{y' \in \mathcal{Y}} \exp(\theta^\top \phi_{y',x})}$. Note that $p_{y,x} \geq 0$ for all $x, y$, and $\sum_{y \in \mathcal{Y}} p_{y,x} = 1$ for all $x$. Then we can have

$$v^\top \left( \sum_{y' \in \mathcal{Y}} \frac{\phi_{y',x} \phi_{y',x}^\top \exp(\theta^\top \phi_{y',x})}{\sum_{y' \in \mathcal{Y}} \exp(\theta^\top \phi_{y',x})} \right.$$

$$\left. - \left( \frac{\sum_{y' \in \mathcal{Y}} \phi_{y',x} \exp(\theta^\top \phi_{y',x})}{\sum_{y' \in \mathcal{Y}} \exp(\theta^\top \phi_{y',x})} \right) \left( \frac{\sum_{y' \in \mathcal{Y}} \phi_{y',x} \exp(\theta^\top \phi_{y',x})}{\sum_{y' \in \mathcal{Y}} \exp(\theta^\top \phi_{y',x})} \right)^\top \right) v$$

$$= \sum_{y' \in \mathcal{Y}} (v^\top \phi_{y',x})^2 p_{y',x} - \left( \sum_{y' \in \mathcal{Y}} v^\top \phi_{y',x} p_{y',x} \right)^2 \geq 0, \quad (32)$$

where the last inequality is due to Jensen's inequality. This suggests that $\nabla_\theta g_{\mathrm{SFT}}(\theta; x, y) \succeq 0$, and the convexity of of $f_{\mathrm{SFT}}$ follows from the fact that

$$\nabla^2 f_{\mathrm{SFT}}(\theta) = \mathbb{E}_{x,y \sim \mathcal{D}_{\mathrm{SFT}}} \left[ \nabla_\theta g_{\mathrm{SFT}}(\theta; x, y) \right]. \quad (33)$$

Note that when $f_{\mathrm{DPO}}$ and $f_{\mathrm{SFT}}$ are convex, $f_{\mathrm{Mix},\lambda}$ is also convex for all $\lambda \in [0, 1]$. $\qquad \square$

### A.2 PROOF OF THEOREM 3.1

*Proof.* For deriving the lower bound given in 3.1, we consider the following problem setup that satisfies Assumption 3.1. Let $\Theta$ be $\mathbb{R}$, and let $\mathcal{D}_{\mathrm{DPO}} = \{(x_1, y_w, y_\ell)\}$ and $\mathcal{D}_{\mathrm{SFT}} = \{(x_2, y)\}$ and consider only two possible outputs exist (i.e. binary classification) such that we have the specification in Table 2. Note that the data point $y'$ is not used in training explicitly, and it is specified to enable the calculation of the output of $\pi_\theta$ with softmax parameterization in the SFT optimization phase. Based on this dataset, we can also define the dataset for reference policy objective as $\mathcal{D}_{\mathrm{ref}} = \{(x_1, y_w)\}$,

| Input | Output | Feature $\phi_{y,x}$ |
|:-----:|:------:|:--------------------:|
| $x_1$ | $y_w = 1$ | $-1.0$ |
| $x_1$ | $y_\ell = 0$ | $-0.5$ |
| $x_2$ | $y = 0$ | $1.0$ |
| $x_2$ | $y' = 1$ | $0.5$ |

Table 2: Data set specification for lower bound analysis example

which has a similar optimization procedure as SFT. Before moving forward, we have to choose the reference policy $\pi_{\text{ref}}$, or equivalently $\theta_{\text{ref}}$. The objective to determine $\theta_{\text{ref}}$ is given by

$$\theta_{\text{ref}} \in \arg\min_\theta f_{\text{ref}}(\theta) := -\log \pi_\theta(y_w \mid x_1), \tag{34}$$

which is graphically illustrated in Figure 5. We choose $\theta_{\text{ref}} = -5$ since this choice reasonably optimizes the objective.

With this problem setup and choice of $\theta_{\text{ref}}$, we can then derive the objectives and corresponding gradients using (1), (4), (23), and (27) as

$$f_{\text{DPO}}(\theta) = \log\left(1 + c\frac{1 + \exp(\theta/2)}{1 + \exp(-\theta/2)}\right) \tag{35}$$

$$f_{\text{SFT}}(\theta) = \log(1 + \exp(-\theta/2)) \tag{36}$$

$$g_{\text{DPO}}(\theta) = \frac{1}{2} \cdot \frac{1}{1 + \exp(-(\theta/2 + 5))} \tag{37}$$

$$g_{\text{SFT}}(\theta) = -\frac{1}{2} \cdot \frac{1}{1 + \exp(\theta/2)}, \tag{38}$$

where $c = \frac{1+\exp(5)}{1+\exp(-5)}$. Choosing $\lambda = 0.5$, we can numerically find an upper bound to $f^*_{\text{Mix},\lambda}$ such that $f^*_{\text{Mix},\lambda} \leq \frac{1}{2}\log c^*$ where $c^* = 173.78$. With this, we can derive a lower bound to $G_{\text{Mix},\lambda}(\theta)$ as

$$
\begin{aligned}
G_{\text{Mix},\lambda}(\theta) &= f_{\text{Mix},\lambda}(\theta) - f^*_{\text{Mix},\lambda} \\
&\geq \frac{1}{2} \cdot \log\left(1 + c\frac{1 + \exp(\theta/2)}{1 + \exp(-\theta/2)}\right) + \frac{1}{2} \cdot \log(1 + \exp(-\theta/2)) - \frac{1}{2}\log c^* \\
&= \frac{1}{2}\log\left(\frac{1}{c^*}(1 + \exp(-\theta/2)) + \frac{c}{c^*}(1 + \exp(\theta/2))\right).
\end{aligned}
\tag{39}
$$

When $\theta = 0$, the right hand side of (39) approximately equals 0.256843. Since it is monotonically increasing for $\theta \in [0, \infty)$, we have $G_{\text{Mix},\lambda}(\theta) \gtrsim 0.256843 = \Omega(1)$ when $\theta \geq 0$. Thus to prove the result, it is sufficient to show Algorithm 1 will generate a $\hat{\theta}_{\text{Seq}}$ that is greater than 0.

We have the first stage iterates:

$$\theta^1_{t+1} = \theta^1_t - \alpha_t \frac{1}{1 + \exp(-\frac{\theta^1_t}{2} - 5)}, \quad \text{for } t = 1, ..., T-1. \tag{40}$$

Using the first stage's last iterate as initial point, we have the second stage iterates:

$$\theta^{2,T}_{t+1} = \theta^{2,T}_t + \alpha_t \frac{1}{1 + \exp(\frac{\theta^{2,T}_t}{2})}, \quad \text{for } t = 1, ..., T-1. \tag{41}$$

where $\theta^{2,T}_1 = \theta^1_T$ and the superscript $T$ in $\theta^{2,T}_t$ indicates the max iteration index in the first stage.

Without loss of generality, we initialize $\theta^1_1 = 0$. Then by (40) and (41), we have

$$\hat{\theta}_{\text{Seq}} = \theta^{2,T}_T = -\alpha_t \sum_{t=1}^{T-1} \frac{1}{1 + \exp(-\frac{\theta^1_t + 10}{2})} + \alpha_t \sum_{t=1}^{T-1} \frac{1}{1 + \exp(\frac{\theta^{2,T}_t}{2})} \tag{42}$$

We first prove the following lemma.

**Lemma A.1.** *Given a positive constant $c$ and a sequence $\{\alpha_t\}$, consider the iterates generated by*

$$\theta_{t+1} = \theta_t + \alpha_t \frac{1}{1 + \exp(c\theta_t)}, \quad \theta'_{t+1} = \theta'_t + \alpha_t \frac{1}{1 + \exp(c\theta'_t)} \tag{43}$$

*If $\theta_1 - \theta'_1 \geq 0$ and $\frac{c\alpha_t}{4} \leq 1$ for any t, then we have*

$$\theta_t - \theta'_t \geq (\theta_1 - \theta'_1)\Pi_{i=1}^{t-1}\left(1 - \frac{c\alpha_i}{4}\right), \quad \forall t.$$

*Proof.* We prove the result by induction. Assume $\theta_t - \theta'_t \geq 0$ for some $t$. We first have

$$\theta_{t+1} - \theta'_{t+1} = \theta_t - \theta'_t + \alpha_t\left(\frac{1}{1 + \exp(c\theta_t)} - \frac{1}{1 + \exp(c\theta'_t)}\right) \tag{44}$$

With $\nabla_\theta \frac{1}{1+\exp(c\theta)} = -c\sigma(-c\bar{\theta})\left(1 - \sigma(-c\theta)\right)$ where $\sigma$ is the sigmoid function, using the mean value theorem in (44), we have for some $\bar{\theta}_t$ in between $\theta_t$ and $\theta'_t$ that

$$\theta_{t+1} - \theta'_{t+1} = \theta_t - \theta'_t - c\alpha_t\sigma(-c\bar{\theta}_t)\left(1 - \sigma(-c\bar{\theta}_t)\right)(\theta_t - \theta'_t)$$

$$\geq \theta_t - \theta'_t - \frac{1}{4}c\alpha_t(\theta_t - \theta'_t)$$

$$= \left(1 - \frac{1}{4}c\alpha_t\right)(\theta_t - \theta'_t)$$

where the inequality follows from $\sigma(-c\bar{\theta}_t)\left(1 - \sigma(-c\bar{\theta}_t)\right) \leq \max_{x\in[0,1]} x(1 - x) = \frac{1}{4}$ and the assumption that $\theta_t - \theta'_t \geq 0$. Since $\theta_1 - \theta'_1 \geq 0$, it follows from induction that $\theta_t - \theta'_t \geq 0$ for any $t$. Then recursively applying the last inequality completes the proof. □

Now we consider (42). Rewriting (40) gives that $\theta_t^1$ is generated by

$$-(\theta_{t+1}^1 + 10) = -(\theta_t^1 + 10) + \alpha_t \frac{1}{1 + \exp(-\frac{\theta_t^1+10}{2})}, \quad \text{for } t = 1, ..., T-1,$$

and $\theta_t^{2,T}$ is generated by (41). Thus $\theta_t^{2,T}$ and $-(\theta_t^1 + 10)$ are generated by (43) with $c = 1/2$. Assuming the step size $\alpha_t$ is proper that $\sum_{t=1}^\infty \alpha_t = \infty$, then there always exists $T^*$ that for $T \geq T^*$, we have $\theta_T^1$ is small enough such that $-(\theta_1^1 + 10) - \theta_1^{2,T} = -10 - \theta_T^1 \geq 0$. If $\alpha_t/8 \leq 1$, we have Lemma A.1 holds for sequences $\theta_t^{2,T}$ and $-(\theta_{t+1}^1 + 10)$, and we have

$$-(\theta_t^1 + 10) \geq \theta_t^{2,T} \Rightarrow \frac{1}{1 + \exp(\theta_t^{2,T}/2)} - \frac{1}{1 + \exp(-(\theta_t^1 + 10)/2)} \geq 0. \tag{45}$$

Using (45) in (42), we have $\hat{\theta}_{\text{Seq}} \geq 0$. This completes the proof. □

### A.3 PROOF OF THEOREM 4.1

In this section, we provide the proof for Theorem 4.1. First, we provide some useful results that will later be used in the main proof. For conciseness, we denote $g_{\text{DPO}}(\theta_t; x^t, y_w^t, y_\ell^t)$ and $g_{\text{SFT}}(\theta_t; x^t, y^t)$ by $g_{\text{DPO},t}$ and $g_{\text{SFT},t}$, respectively.

**Lemma A.2** ((Rockafellar, 1970) Theorem 25.1 and Corollary 25.11). *Consider any convex differentiable function $f$ and let $\theta \in \Theta$. Then, for any $\theta' \in \Theta$, we have*

$$f(\theta') \geq f(\theta) + \nabla f(\theta)^\top(\theta' - \theta). \tag{46}$$

**Lemma A.3** ((Orabona, 2020) Lemma 1.). *Let $\{\eta_t\}_{t=1}^T$ be a non-increasing sequence of positive numbers and $q_t \geq 0$ for all $t = 1, \ldots, T$. Then*

$$n_T q_T \leq \frac{1}{T}\sum_{t=1}^T \eta_t q_t + \sum_{k=1}^{T-1} \frac{1}{k(k+1)} \sum_{t=T-k+1}^T \eta_t(q_t - q_{T-k}) \tag{47}$$

**Lemma A.4.** *Consider iterates $\theta_t$ and $\theta_{t+1}$ generated by Algorithm 2 for $t \in \{1, \ldots, T-1\}$. Then for any $\theta' \in \Theta$ and $\lambda \in [0, 1]$, we have*

$$\mathbb{E}\left[f_{Mix,\lambda}(\theta_t) - f_{Mix,\lambda}(\theta')\right] \leq \frac{1}{2\alpha_t}\mathbb{E}\left[\|\theta_t - \theta'\|^2\right] - \frac{1}{2\alpha_t}\mathbb{E}\left[\|\theta_{t+1} - \theta'\|^2\right] + \frac{\alpha_t}{2}M_\lambda^2, \quad (48)$$

*where $M_\lambda = \lambda M_1 + (1 - \lambda)M_2$, and $M_1, M_2$ are as defined in Proposition A.1.*

*Proof.* Considering Algorothm 2, for any $\theta' \in \Theta$, we can have

$$\|\theta_{t+1} - \theta'\|^2 - \|\theta_t - \theta'\|^2 \leq \|\theta_t - \alpha_t(\mathbb{I}_{i_t=1}g_{\mathrm{DPO},t} + \mathbb{I}_{i_t=0}g_{\mathrm{SFT},t}) - \theta'\|^2 - \|\theta_t - \theta'\|^2$$

$$= -2\alpha_t(\mathbb{I}_{i_t=1}g_{\mathrm{DPO},t} + \mathbb{I}_{i_t=0}g_{\mathrm{SFT},t})^\top(\theta_t - \theta')$$

$$+ \alpha_t^2\|\mathbb{I}_{i_t=1}g_{\mathrm{DPO},t} + \mathbb{I}_{i_t=0}g_{\mathrm{SFT},t}\|^2$$

$$= -2\alpha_t(\mathbb{I}_{i_t=1}g_{\mathrm{DPO},t} + \mathbb{I}_{i_t=0}g_{\mathrm{SFT},t})^\top(\theta_t - \theta')$$

$$+ \alpha_t^2\left(\mathbb{I}_{i_t=1}\|g_{\mathrm{DPO},t}\|^2 + \mathbb{I}_{i_t=0}\|g_{\mathrm{SFT},t}\|^2\right) \quad (49)$$

Taking conditional expectation $\mathbb{E}[\cdot \mid \theta_t]$ (over randomness of datapoints and $i_t$) in both sides of above inequality, we obtain

$$\mathbb{E}\left[\|\theta_{t+1} - \theta'\|^2 \mid \theta_t\right] - \|\theta_t - \theta'\|^2 \leq -2\alpha_t(\lambda\nabla f_{\mathrm{DPO}}(\theta_t) + (1 - \lambda)\nabla f_{\mathrm{SFT}}(\theta))^\top(\theta_t - \theta') + \alpha_t^2 M_\lambda^2$$

$$= -2\alpha_t\nabla f_{\mathrm{Mix},\lambda}(\theta_t)^\top(\theta_t - \theta') + \alpha_t^2 M_\lambda^2$$

$$\leq -2\alpha_t\left(f_{\mathrm{Mix},\lambda}(\theta_t) - f_{\mathrm{Mix},\lambda}(\theta')\right) + \alpha_t^2 M_\lambda^2, \quad (50)$$

where the first inequality is using the definitions of $g_{\mathrm{DPO},t}, g_{\mathrm{SFT},t}$, and $M_\lambda$, equality is by the defintion of $f_{\mathrm{Mix},\lambda}$, and the last inequality is due to Lemma A.2. The result follows from taking total expectation in both sides and rearranging the inequality (50). $\square$

With the above results, we are ready to prove Theorem 4.1.

**Theorem A.1** (Theorem 4.1 Restated with Additional Details). *Consider Algorithm 2 with $\alpha_t = \alpha_0/\sqrt{T}$ for all $t \in \{1, \ldots, T\}$ and $\alpha_0 > 0$. Then, under Assumption 3.1, for any $\lambda \in [0, 1]$, we have*

$$\mathbb{E}[G_{Mix,\lambda}(\theta_T)] \leq \frac{\alpha_0}{2\sqrt{T}}\|\theta_1 - \theta^*_{Mix,\lambda}\|^2 + \frac{(2 + \log(T-1))M_\lambda^2\alpha_0}{2\sqrt{T}} \quad (51)$$

*where $\theta^*_{Mix,\lambda} \in \arg\min_{\theta \in \Theta} f_{Mix,\lambda}(\theta)$, and $M_\lambda = \lambda M_1 + (1 - \lambda)M_2$ with $M_1, M_2$ as defined in Assumption A.1.*

*Proof.* Substituting $\eta_t = \alpha$ and $q_t = G_{\mathrm{Mix},\lambda}(\theta_t)$ in (47) (Lemma A.3), we have

$$G_{\mathrm{Mix},\lambda}(\theta_T) \leq \frac{1}{T}\sum_{t=1}^T G_{\mathrm{Mix},\lambda}(\theta_t) + \sum_{k=1}^{T-1}\frac{1}{k(k+1)}\sum_{t=T-k+1}^T (G_{\mathrm{Mix},\lambda}(\theta_t) - G_{\mathrm{Mix},\lambda}(\theta_{T-k}))$$

$$= \frac{1}{T}\sum_{t=1}^T G_{\mathrm{Mix},\lambda}(\theta_t) + \sum_{k=1}^{T-1}\frac{1}{k(k+1)}\sum_{t=T-k+1}^T (f_{\mathrm{Mix},\lambda}(\theta_t) - f_{\mathrm{Mix},\lambda}(\theta_{T-k})), \quad (52)$$

where we have used the definition of $G_{\mathrm{Mix},\lambda}$ in the equality. Taking total expectation on both sides of (52), we get

$$\mathbb{E}[G_{\mathrm{Mix},\lambda}(\theta_T)] \leq \frac{1}{T}\sum_{t=1}^T \mathbb{E}[G_{\mathrm{Mix},\lambda}(\theta_t)] + \sum_{k=1}^{T-1}\frac{1}{k(k+1)}\sum_{t=T-k+1}^T \mathbb{E}\left[f_{\mathrm{Mix},\lambda}(\theta_t) - f_{\mathrm{Mix},\lambda}(\theta_{T-k})\right] \quad (53)$$

The first term in the right hand side of (53) can be bounded by choosing $\theta' = \theta^*_{\mathrm{Mix},\lambda}$ and $\alpha_t = \alpha$ in (48) from Lemma A.4 and then taking a telescoping sum:

$$\sum_{t=1}^T \mathbb{E}[G_{\mathrm{Mix},\lambda}(\theta_t)] \leq \frac{1}{2\alpha}\|\theta_1 - \theta^*_{\mathrm{Mix},\lambda}\|^2 - \frac{1}{2\alpha}\|\theta_{T+1} - \theta^*_{\mathrm{Mix},\lambda}\|^2 + T\frac{M_\lambda^2\alpha}{2}$$

$$\leq \frac{1}{2\alpha}\|\theta_1 - \theta^*_{\mathrm{Mix},\lambda}\|^2 + T\frac{M_\lambda^2\alpha}{2} \quad (54)$$

---

**Algorithm 4** Memory Efficient MAXRIGHT

---

1: Input $\mathcal{D}_{\text{DPO}}, \mathcal{D}_{\text{SFT}}, \{\alpha_t\}_{t=1}^T, \lambda \in [0,1]$, max evaluation steps $k$
2: Initialize $\theta_1 \in \Theta$
3: **for** $t = 1, \ldots, T-1$ **do**
4:     Sample $x_1^t, y_w^t, y_\ell^t \sim \mathcal{D}_{\text{DPO}}$
5:     Sample $x_2^t, y^t \sim \mathcal{D}_{\text{SFT}}$
6:     **if** $t \bmod k = 0 \,\|\, t = 1$ **then**
7:         Evaluate (without generating computational graph)
            $\bar{f}_{1,\lambda}(\theta_t) := \lambda\left(f_{\text{DPO}}(\theta_t; x_1^t, y_w^t, y_\ell^t) - f_{\text{DPO}}^*\right)$ and
            $\bar{f}_{2,\lambda}(\theta_t) := (1-\lambda)\left(f_{\text{SFT}}(\theta_t; x_2^t, y^t) - f_{\text{SFT}}^*\right)$
8:         Set
            $t_0 = t$
            $\bar{f}_{1,\lambda}^{\text{stale},t_0} = \bar{f}_{1,\lambda}(\theta_t)$
            $\bar{f}_{2,\lambda}^{\text{stale},t_0} = \bar{f}_{2,\lambda}(\theta_t)$
9:     **end if**
10:    Set $i_t = \arg\max_i \bar{f}_{1,\lambda}^{\text{stale},t_0}$
11:    **if** $i_t = 1$ **then**
12:       Set $\bar{f}_{1,\lambda}^{\text{stale},t_0} = \lambda\left(f_{\text{DPO}}(\theta_t; x_1^t, y_w^t, y_\ell^t) - f_{\text{DPO}}^*\right)$
13:       Update $\theta_{t+1} = \Pi_\Theta\left(\theta_t - \alpha_t g_{\text{DPO}}(\theta_t; x_1^t, y_w^t, y_\ell^t)\right)$
14:    **else**
15:       Set $\bar{f}_{2,\lambda}^{\text{stale},t_0} = (1-\lambda)\left(f_{\text{SFT}}(\theta_t; x_2^t, y^t) - f_{\text{SFT}}^*\right)$
16:       Update $\theta_{t+1} = \Pi_\Theta\left(\theta_t - \alpha_t g_{\text{SFT}}(\theta_t; x_2^t, y^t)\right)$
17:    **end if**
18: **end for**
19: Output $\hat{\theta}_{\text{MAX}} := \theta_T$

---

Now we consider the second term in the right hand side of (53). We first have

$$\sum_{t=T-k+1}^{T} \mathbb{E}\left[f_{\text{Mix},\lambda}(\theta_t) - f_{\text{Mix},\lambda}(\theta_{T-k})\right] = \sum_{t=T-k}^{T} \mathbb{E}\left[f_{\text{Mix},\lambda}(\theta_t) - f_{\text{Mix},\lambda}(\theta_{T-k})\right]$$

$$\leq (k+1)\frac{M_\lambda^2 \alpha}{2} \tag{55}$$

where the inequality follows from setting $\theta' = \theta_{T-k}$, $\alpha_t = \alpha$ in (48) from Lemma A.4 and then taking a telescoping sum from $t = T - k$ to $T$. Substituting the last inequality to the second term in the right hand side of (53) yields

$$\sum_{k=1}^{T-1} \frac{1}{k(k+1)} \sum_{t=T-k+1}^{T} \mathbb{E}\left[f_{\text{Mix},\lambda}(\theta_t) - f_{\text{Mix},\lambda}(\theta_{T-k})\right] \leq \sum_{k=1}^{T-1} \frac{1}{k}\frac{M_\lambda^2 \alpha}{2}$$

$$\leq (1 + \log(T-1))\frac{M_\lambda^2 \alpha}{2}. \tag{56}$$

Choosing $\alpha = \frac{\alpha_0}{\sqrt{T}}$, and substituting (54) and (56) in (53) yields

$$\mathbb{E}[G_{\text{Mix},\lambda}(\theta_T)] \leq \frac{\alpha_0}{2\sqrt{T}}\|\theta_1 - \theta_{\text{Mix},\lambda}^*\|^2 + \frac{(2 + \log(T-1))M_\lambda^2 \alpha_0}{2\sqrt{T}}$$

which completes the proof. $\qquad\square$

## B  MEMORY EFFICIENT MAXRIGHT IMPLEMENTATION.

In this section we summarize the memory-efficient implementation of MAXRIGHT in Section 4.2.

## C  ADDITIONAL EXPERIMENT RESULTS

In this section, we provide additional experiment results using PYTHIA-1B and LLAMA3-8B models.

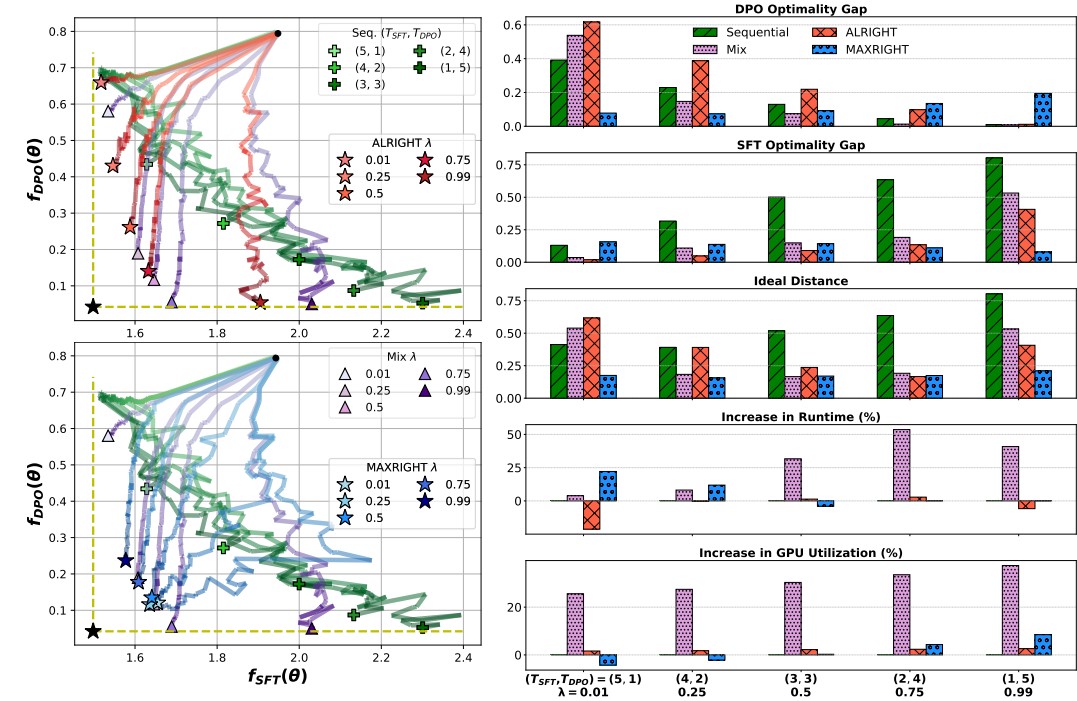

Figure 6: Comparison of proposed methods with baselines (SFT first then DPO for Sequential). **Left:** Training trajectories for various methods in the objective space, visualizing their convergence with respect to the DPO and SFT objectives. **Right:** Performance comparison across multiple evaluation metrics, including optimality gap for DPO and SFT objectives, ideal distance, runtime, and GPU utilization. The bar charts highlight the trade-offs and resource efficiency of each method for different choices of $(T_{\text{SFT}}, T_{\text{DPO}})$ or $\lambda$.

C.1 ADDITIONAL EXPERIMENTS WITH PYTHIA-1B

**ALRIGHT and MAXRIGHT significantly outperform Sequential.** In Figure 6 (left), it can be seen that the final models obtained by ALRIGHT and MAXRIGHT achieve better trade-off in DPO and SFT objective values in general compared to Sequential. Furthermore, ALRIGHT and MAXRIGHT perform comparably or significantly better in terms of SFT optimality gap and ideal distance metrics (Figure 6 (right)), while Sequential demonstrates a better performance in RLHF optimality gap. This is because, in this experiment setup, Sequential is implemented by optimizing for SFT first then DPO.

**ALRIGHT and MAXRIGHT require minimal additional resources compared to Sequential and significantly lower than Mix.** In Figure 6 (right), the additional computational resources required by different implementations of ALRIGHT and MAXRIGHT are minimal (or even negative) relative to their Sequential counterparts. In contrast, Mix incurs substantial additional resource usage, with increases of upto $53\%$ in runtime and upto $37\%$ in GPU utilization, despite achieving comparable performance metrics to ALRIGHT and MAXRIGHT.

C.2 ADDITIONAL EXPERIMENTS WITH LLAMA3-8B

**ALRIGHT and MAXRIGHT perform comparably or better than Sequential.** In Figure 7 (left), it can be seen that the final models obtained by ALRIGHT and MAXRIGHT achieve better or comparable trade-off in DPO and SFT objective values in general compared to Sequential. Furthermore, MAXRIGHT performs consistently better in terms of ideal distance metric (Figure 7 (right)), which is consistent with PYTHIA-1B experiments.

**ALRIGHT and MAXRIGHT require minimal additional resources compared to Sequential and significantly lower than Mix.** In Figure 7 (right), the additional computational resources required by

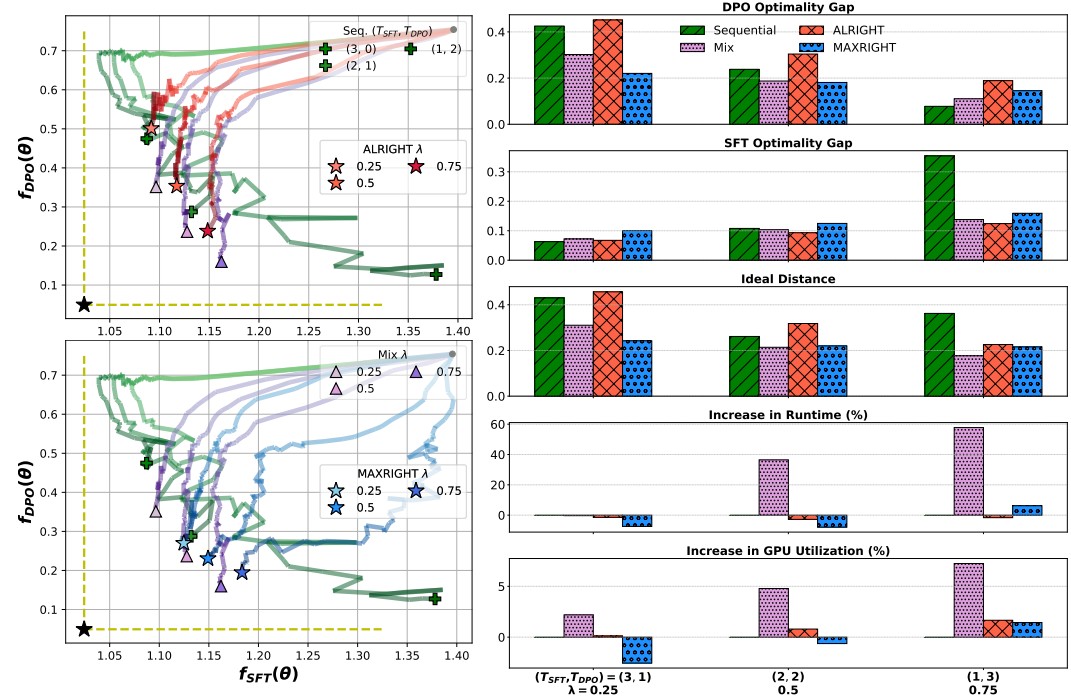

Figure 7: Comparison of proposed methods with baselines (SFT first then DPO for Sequential) using LLAMA3-8B. **Left:** Training trajectories for various methods in the objective space, visualizing their convergence with respect to the DPO and SFT objectives. **Right:** Performance comparison across multiple evaluation metrics, including optimality gap for DPO and SFT objectives, ideal distance, runtime, and GPU utilization. The bar charts highlight the trade-offs and resource efficiency of each method for different choices of $(T_{\text{SFT}}, T_{\text{DPO}})$ or $\lambda$.

different implementations of ALRIGHT and MAXRIGHT are minimal (or even negative) relative to their Sequential counterparts. In contrast, Mix incurs substantial additional resource usage, with increases of upto $58\%$ in runtime and upto $7\%$ in GPU utilization, despite achieving comparable performance metrics to ALRIGHT and MAXRIGHT. Note that, unlike in PYTHIA-1B experiments, the increase in GPU utilization for Mix is lower. We believe this is because we implement gradient checkpointing when training LLAMA3-8B, and gradient checkpointing improves GPU utilization at the cost of increased runtime due to duplicate activation computations in the backward pass.

## D    EXPERIMENT DETAILS

In this section, we provide experiment details for the experiments in Sections 6 and C. We build upon OPENRLHF (Hu et al., 2024) framework to implement the experiments in Section 6, C.1, and C.2.

### D.1    EXPERIMENTS DETAILS FOR TOY ILLUSTRATION IN FIGURE 2

In this section we provide details for the experiment results given in Figure 2. We consider $\Theta$ be $\mathbb{R}^2, \mathcal{D}_{\text{DPO}} = \{(x_1, y_w, y_\ell)\}$ and $\mathcal{D}_{\text{SFT}} = \{(x_2, y)\}$ and setting where only two possible outputs exist (i.e. binary classification) such that we have the specification in Table 2. Note that the data point $y'$ is not used in training explicitly, and it is specified to enable the calculation of the output of $\pi_\theta$ with softmax parameterization in the SFT optimization phase. Based on this dataset, we can also define the dataset for reference policy objective as $\mathcal{D}_{\text{ref}} = \{(x_1, y_w)\}$, which has a similar optimization procedure as SFT.

To obtain $\theta_{\text{ref}}$, we train a parameter initialized at $[5.0; -9.9]^\top$ for 1000 epochs with a learning rate of 0.01. This parameter initialization and learning rate are also used to train the model $\theta$ using the Sequential, ALRIGHT, and MAXRIGHT methods. Furthermore, for illustration purposes, we use a

weight decay of $0.001$ in optimization. The resulting $\pi_{\mathrm{ref}}$ is then used as the reference policy for the DPO objective in all three methods.

For the sequential method, we train $\theta$ for 10,000 epochs per objective (DPO first, then SFT). A threshold of 0.05 for the objective value is applied to stop training for a given objective, preventing excessive overfitting to that objective.

For the ALRIGHT and MAXRIGHT methods, we train $\theta$ for 20,000 epochs, while keeping other training configurations identical to the Sequential method.

| Input | Output | Feature $\phi_{y,x}$ |
|-------|--------|----------------------|
| $x_1$ | $y_w = 1$ | $[1.0; 1.0]^\top$ |
| $x_1$ | $y_\ell = 0$ | $[0.5; 0.5]^\top$ |
| $x_2$ | $y = 0$ | $[1.0; 0.5]^\top$ |
| $x_2$ | $y' = 1$ | $[0.5; 0.5]^\top$ |

Table 3: Data set specification for toy illustration in Figure 2

### D.2 ADDITIONAL DETAILS FOR EXPERIMENTS WITH PYTHIA-1B

We conducted three sets of experiments using the PYTHIA-1B model:

(1) Comparison of baselines and proposed methods for sequential training with DPO first, followed by SFT (Figure 3),

(2) Comparison of baselines and proposed methods for sequential training with SFT first, followed by DPO (Figure 6), and

(3) Ablation study on the choice of maximum evaluation steps for memory-efficient MAXRIGHT (Figure 4). The primary difference between the first two experiments is the $\pi_{\mathrm{ref}}$ used (and thus the DPO objective), as described in Section 2.

For training the models (both $\theta$ and $\theta_{\mathrm{ref}}$) in experiments (1), (2), and (3), we use LoRA with rank 32 and $\alpha = 32$. The QUERY_KEY_VALUE layers are the target modules to which LoRA is applied. No gradient checkpointing is used for PYTHIA-1B training. The learning rate is set to $5 \times 10^{-5}$ for all model training with PYTHIA-1B.

To obtain $\theta_{\mathrm{ref}}$ for experiments (1) and (3), we train the model for 6 epochs using $24,000$ input-response pairs from the RM-HH-RLHF dataset, with a batch size of 12 and a learning rate of $5 \times 10^{-5}$. For experiment (2), we train the model for 6 epochs using $24,000$ samples from the ALPACA-GPT4 dataset, with a batch size of 24.

To compute $f_{\mathrm{DPO}}^*$ and $f_{\mathrm{SFT}}^*$, which are required for calculating the optimality gap, ideal distance metrics, and implementing the MAXRIGHT method, we perform independent optimization of the DPO and SFT objectives for 6 epochs. For the SFT objective, we use $24,000$ samples from the ALPACA-GPT4 dataset with a batch size of 24, and for the DPO objective, we use $8,000$ samples from the RM-HH-RLHF dataset with a batch size of 8. Additionally, we run ALRIGHT for 6 epochs to establish a reference Pareto front, which, along with the optimal objective values, is used as a stopping criterion for joint optimization. No stopping criterion is applied for the sequential method.

Finally, all methods are trained for 6 epochs, using the corresponding $\lambda$ for joint optimization methods or a combination of $T_{\mathrm{DPO}}$ and $T_{\mathrm{SFT}}$ for the sequential method, until the stopping criterion is reached. For the memory-efficient MAXRIGHT implementation in experiments (1) and (2), the maximum evaluation step is set to 10.

### D.3 ADDITIONAL DETAILS FOR EXPERIMENTS WITH LLAMA3-8B

We present the experiment results for LLAMA3-8B training in Table 1 and Figure 7. Both result sets share the same training configuration, which is described below.

For training the models (both $\theta$ and $\theta_{\mathrm{ref}}$), we use LoRA with rank 16 and $\alpha = 16$. The Q_PROJ and V_PROJ layers are the target modules for LoRA application. Gradient checkpointing is enabled

during training with LLAMA3-8B. The learning rate is set to $5 \times 10^{-5}$ for all model training with LLAMA3-8B. To obtain $\theta_{\text{ref}}$, we train the model for 4 epochs using $24,000$ samples from the ALPACA-GPT4 dataset, with a batch size of 16.

To compute $f^*_{\text{DPO}}$ and $f^*_{\text{SFT}}$, required for calculating the optimality gap, ideal distance metrics, and implementing MAXRIGHT, we independently optimize the DPO and SFT objectives for 4 epochs. We use $24,000$ samples from the ALPACA-GPT4 dataset for the SFT objective with a batch size of 16, and $6,000$ samples from the RM-HH-RLHF dataset for the DPO objective with a batch size of 4. Additionally, we run ALRIGHT for 4 epochs to establish a reference Pareto front, which, along with the optimal objective values, serves as a stopping criterion for joint optimization. No stopping criterion is applied for the sequential method.

Finally, all methods are trained for 4 epochs, using the corresponding $\lambda$ for joint optimization methods or a combination of $T_{\text{DPO}}$ and $T_{\text{SFT}}$ for the sequential method, until the stopping criterion is reached. For memory-efficient MAXRIGHT implementation, the maximum evaluation step is set to 10.

### D.4 EVALUATION METRICS USED FOR MEASURING RESOURCE USAGE

In this section we give the formula for computing the resource usage metrics used in Section 6; percentage increase in runtime and percentage increase in GPU utilization.

Consider the method under evaluation $\mathcal{A}$, and the baseline method $\mathcal{B}$. Then, percentage increase in runtime is given by

$$\text{percentage increase in runtime for } \mathcal{A} = \frac{\text{runtime of } \mathcal{A} - \text{runtime of } \mathcal{B}}{\text{runtime of } \mathcal{B}} \times 100\%. \tag{57}$$

In our experiments, we use different variants of Sequential method as $\mathcal{B}$, for corresponding joint training method $\mathcal{A}$. For example, in PYTHIA-1B experiments we use Sequential with $(T_{\text{DPO}}, T_{\text{SFT}}) = (5, 1)$ configuration as the baseline for Mix, ALRIGHT, and MAXRIGHT with $\lambda = 0.99$. We can similarly define the percentage increase of GPU utilization as

$$\text{percentage increase in GPU utilization for } \mathcal{A} = \frac{\text{GPU utilization of } \mathcal{A} - \text{GPU utilization of } \mathcal{B}}{\text{GPU utilization of } \mathcal{B}} \times 100\%. \tag{58}$$

Here, the GPU utilization is computes as the median GPU utilzation throughout the runtime of a given method.

