# OpenReview forum: "Mitigating Forgetting in LLM Supervised Fine-Tuning and Preference Learning"
_ICLR.cc/2025/Conference — Submitted to ICLR 2025_

### Official Review · Reviewer_w7o1 · 2024-10-17

**Soundness:** 2
**Presentation:** 2
**Contribution:** 1
**Rating:** 3
**Confidence:** 4

**Summary:**

This paper first shows that there is a gap between the optimal point of the convex sum of DPO and SFT since iteratively minimizing SFT and DPO will make the model osillate. Although optimizing a linear combinarion of DPO and SFT can be a solution, it is computationally intensive. To solve this computation issue, this paper suggests (1) sampling objective function with bernouilli distribution or (2) minimize the maximum loss between SFT and DPO.

**Strengths:**

- This paper tries to explain what it suggests theoretically.

**Weaknesses:**

- Authors said it should be pareto-optimal between SFT and DPO. Does the method suggested by rthe authors prove pareto optimality?
- No Standard deviations for figure 1. I cannot trust its confidence.
- What figures mean are not clear. For example, how can we recognize where black stars (=ideal point) are located? Are x axis and y axis are loss empicial loss values? How the training trajectories can be drawn? It means every gradient update? or every epoch? Which seed? For figure 3 and 4, bar color and legend does not match.
- Also with no standard deviation, I cannot evaluate whether the test result is statistically meaningful or not.

**Questions:**

- Why should we find a point that optimizes both SFT and DPO? Finding a point which optimizes DPO loss seems enough..
- More specifically, if optimizing DPO causes negative impact to SFT, it means that SFT fails to align its result to human preference, which is not a desirable situation. Then, why should we consider optimizing the unwanted SFT also?
- There are papers which do not require SFT and Preference Optimization phase, rather, does both at once, e.g. SimPO[1] or ORPO[2]. Those papers would not suffer from iterative update and oscillations. What will be the difference with those papers and yours and what is the value of this paper?

[1] Meng, Y., Xia, M., & Chen, D. (2024). Simpo: Simple preference optimization with a reference-free reward. arXiv preprint arXiv:2405.14734.

[2] Hong, J., Lee, N., & Thorne, J. (2024). Reference-free monolithic preference optimization with odds ratio. arXiv preprint arXiv:2403.07691.

---

### Official Review · Reviewer_WBeQ · 2024-11-01

**Soundness:** 3
**Presentation:** 2
**Contribution:** 3
**Rating:** 8
**Confidence:** 4

**Summary:**

In this paper, the authors provide a thorough analysis on the suboptimality of the traditional 2-stage process of preference learning and supervised fine-tunining for LLM training. It is shown that this sub-optimality is due to the incompatibility between the objectives in preference learning and supervised fine-tuning. The authors then proposed two algorithms (ALRIGHT, and MAXRIGHT) that treat this problem as multi-objective learning. Both of these algorithms achieve the Pareto optimal solution for multi-objective problem. Experiments show that the proposed algorithms requires little increase in runtime and GPU utilization. The proposed algorithms also achieve large improvement in both SFT and DPO objectives.

**Strengths:**

- Originality
The problem concerned in this paper is novel as the SFT and RLHF are industrial standard in LLM training, but not many discussion on the joint suboptimality of the entire process. This paper deepens the understanding on the mismatch between labeling logic in training data and additional labeling logic from human preference or RL agent. The proposal which treats this problem as a multi-objective learning is novel as well.

- Quality
The proof looks correct based on my brief review. The visualization of suboptimality of sequential two-stage process in Figure 2 is interesting. The experiments which indicates that the proposed method is able to reach Pareto front of multi-objective problem is convincing.

**Weaknesses:**

Clarity
- The figures 2,3,4 when compared the trajectory of learning with respect to two objectives are not clear. It is due to the gradual changing color of the line and inconsistent line and color pattern. In all, these figures need some rework. It is suggested that simple dash/solid line with different marker would be sufficient to tell apart different settings. It is also important to note that this paper heavily rely on color print to tell the result. Under gray-scale printout, these figures are not readable.
- Figure 2 legend is too large and covers the details of the contour plot.

Suggestions:
- Use distinct line styles (solid, dashed, dotted) and markers (circles, squares, triangles) for different methods.
Implement a consistent color scheme across all figures.
- Add labels directly to the lines rather than relying solely on a legend.
- Include alternative visualizations (e.g., separate plots) or annotations that don't rely solely on color, to ensure the key information is accessible in grayscale as well.

**Questions:**

- Theorem 4.1 is for randomly selection of DPO and SFT objectives, while Theorem 3.1. is for sequential training with DPO and SFT. But the Left-hand-side is similar, it is unclear from the theorem statement which part has changed. It is suggested that some modification on the theorem 4.1. LHS of the bound so that we do not confuse the result.

- How this approach resolve the conflicts in labeling logic between DPO and SFT? One way to mitigate the forgetting issue is to increase the label diversity and increase the label consistency between training data in SFT and training data in DPO. It is unclear if random sampling or multi-objective learning would help to mitigate the inconsistency of data between DPO and SFT.

- Did we have a good understanding on what proportion of time for SFT vs. DPO would be more efficient ?

- One issue for multi-objective learning is that we assume every solution in Pareto front is admissible for the solution, but in practice, the performance of LLM is judged by metrics other than that of DPO and SFT. I am wondering if every solution in Pareto front would truely equivalent from user perspective.

---

### Official Review · Reviewer_BAvq · 2024-11-04

**Soundness:** 3
**Presentation:** 3
**Contribution:** 3
**Rating:** 6
**Confidence:** 3

**Summary:**

This paper investigates the forgetting issue during the post-training of large language models (LLMs). Specifically, it first identifies and examines the problem of forgetting and suboptimal performance in sequential post-training setups, where Direct Preference Optimization (DPO) is followed by Supervised Fine-Tuning (SFT), through both theoretical analysis and empirical study. Based on these findings, the authors propose two efficient algorithms, ALRIGHT and MAXRIGHT, which perform joint post-training, achieving lower loss and improved performance. Experimental results substantiate the effectiveness of the proposed methods.

**Strengths:**

- The paper is well-organized and easy to follow, with a clear motivation. The theoretical contributions and methodologies are logically developed.

- The study of post-training schema and the forgetting issue in LLMs addresses significant and interesting challenges in the field.

- The proposed methods are both effective and efficient, making them highly practical.

**Weaknesses:**

While I have no major concerns, I do have several questions outlined in the Questions section.

**Questions:**

- Theoretical Questions

    - Regarding Theorem 3.1, which assumes $ T_{DPO} = T_{SFT} = T$ : Is this assumption too strong and potentially misaligned with real-world scenarios?

    - Modeling of LLMs: The approach in line 215 differs from practical methods like LoRA used in experiments. This theoretical-experimental gap may somewhat impact the study's contributions.

- Experimental Questions

    - Baseline Comparisons: The authors primarily compare their proposed methods with mixed and sequential post-training settings. Comparing with other approaches that balance the Reinforcement Learning from Human Feedback (RLHF) and SFT processes, as mentioned in related work, would further strengthen the study’s contributions.

- Presentation and Expression

    - Notation Consistency: Unifying the definitions of $\mathcal{D}_{RLHF}$  and $ \mathcal{D}_{DPO}$ would enhance clarity in the Preliminaries section.

---

### Official Review · Reviewer_18Wk · 2024-11-04

**Soundness:** 3
**Presentation:** 3
**Contribution:** 2
**Rating:** 5
**Confidence:** 4

**Summary:**

The paper is motivated by the issue of forgetting in LLM post-training during the supervised fine-tuning (SFT) and direct preference optimization (DPO) stages which are typically applied sequentially. Sequential training, however, leads to "forgetting" of the initial training stage as the model focuses on new objectives. The authors propose a joint post-training framework with two multi-objective optimization methods, ALRIGHT and MAXRIGHT. These methods alternate between the objectives to retain balance without incurring significant additional computational costs. ALRIGHT provides theoretical convergence guarantees, while MAXRIGHT adapts dynamically to optimize both  objectives effectively. Empirical tests on models like LLAMA-3 and PYTHIA-1B demonstrate that these methods improve the trade-off between tasks, reduce forgetting, and enhance performance on benchmarks, achieving up to a 31% higher win rate than sequential training with minimal resource demands.

**Strengths:**

The paper has the following strengths:
* **Keen observation on forgetfulness in LLM post-training:** The paper demonstrates forgetfulness of the sequential method in post-training.
* **Alternation instead of Mixing:** The paper explains why using the "vanilla" version of the linear scalarization might be computationally prohibitive. They introduce the stochastic equivalent of linear scalarization in ALRIGHT.

**Weaknesses:**

The paper has the following weaknesses:
* **Not Novel; already well-known Multi-Objective Optimization Methods**: The paper is simply re-iterating well-known methods of Multi-Objective Optimization (MOO) in the literature: 1) ALRIGHT is simply the stochastic version of "Linear scalarization" and 2) MAXRIGHT is basically the Chebyshev Scalarization.
* **Easy example in Figure 2**: In the toy example provided in Figure 2, one expects to see DPO and SFT pure solutions are Pareto Optimal, however, it seems that they are both dominated by "Average Opt. Point".
* **Apriori knowledge of** $f^*$: The MAXRIGHT method requires the apriori knowledge of $f^*$ for both SFT and DPO. This would basically add to the computational costs which is not discussed.
* **Ideal Distance is scale-dependent:** The "Ideal Distance" that is introduced is scale-dependent and does not represent a "fair" combination of both objectives. One can simply scale up one objective by an arbitrary multiplication and bias the Idea Distance towards either of the objectives.

**Questions:**

I have no questions.

**Details Of Ethics Concerns:**

No ethics concerns

---

### Meta-Review · Area_Chair_KFUc · 2024-12-23

**Metareview:**

This paper presents a new approach to post-training of LLMs using a multi-objective function designed for both supervised fine-tuning (SFT) and preference learning (RLHF or DPO). This is a departure from conventional approaches that perform these two steps sequentially, and suffer from the risk of the LLM forgetting what it learned in the first stage.

This is an interesting idea. The analysis of the suboptimality of the standard post-training method for LLM is also sound. The paper also demonstrates that the multi-objective optimization approach leads to empirical gains as well.

However, the reviewers had a mixed opinion. Some of the reviewers engaged in discussion with the authors and some of the concerns were addressed.

Reviewers 18Wk who initially gave a score of 3 raised the score to 5 during the discussion phase. However, Reviewers 18Wk added that the paper uses standard multi-objective optimization methods and that LLM researchers may already be using similar approaches and thus the paper's contributions may not be surprising to the LLM community. A similar sentiment was also shared by Reviewer w7o1.

Although Reviewers 18Wk and w7o1 didn't point out any specific works that already do this, I would like to point out some recent works that employ multi-objective optimization, such as:

Controllable Preference Optimization: Toward Controllable Multi-Objective Alignment (Guo et al, EMNLP 2024)
Rewards-in-Context: Multi-objective Alignment of Foundation Models with Dynamic Preference Adjustment (Yang et al, 2024)

There were some additional concerns such as need for improved presentation of results (Reviewers 18Wk, BAvq, w7o1) and need for experiments with more seeds and baselines (Reviewers BAvq, w7o1).

In the end, because of these concerns, the paper fell short of the acceptance criteria. The authors are encouraged to look into other recent methods that use multi-objective optimization and reconsider how to position and distinguish their work in light of those methods.

**Additional Comments On Reviewer Discussion:**

The rebuttal was considered by the reviewers and some discussion took place among the authors and reviewers. Reviewers 18Wk who initially gave a score of 3 raised the score to 5 during the discussion phase. However, Reviewers 18Wk added that the paper uses standard multi-objective optimization methods and that LLM researchers may already be using similar approaches and thus the paper's contributions may not be surprising to the LLM community. A similar sentiment was also shared by Reviewer w7o1.

There were some additional concerns such as need for improved presentation of results (Reviewers 18Wk, BAvq, w7o1) and need for experiments with more seeds and baselines (Reviewers BAvq, w7o1).

In the end, the reviewers with these concerns didn't feel adequately satisfied to support the paper for acceptance.

The decision was made after factoring in all these points.

---

### Decision · Program_Chairs · 2025-01-22

Reject